# Stroma-infiltrating T cell spatiotypes define immunotherapy outcomes in adolescent and young adult patients with melanoma

Xinyu Bai [1,2,3], Grace H. Attrill [1,2,3], Tuba N. Gide[1,2,3], Peter M. Ferguson [1,2,4,5], Kazi J. Nahar[1,2,3], Ping Shang[1,2,3], Ismael A. Vergara [1,2,3], Umaimainthan Palendira [1,2,3,6], Ines Pires da Silva[1,2,3,7], Matteo S. Carlino[1,7], Alexander M. Menzies[1,2,8,9], Georgina V. Long [1,2,3,8,9,10], Richard A. Scolyer [1,2,3,4,5,10], James S. Wilmott[1,2,3,10] & Camelia Quek [1,2,3,10] ✉

The biological underpinnings of therapeutic resistance to immune checkpoint inhibitors (ICI) in adolescent and young adult (AYA) melanoma patients are incompletely understood. Here, we characterize the immunogenomic profile and spatial architecture of the tumor microenvironment (TME) in AYA (aged ≤ 30 years) and older adult (aged 31–84 years) patients with melanoma, to determine the AYA-specific features associated with ICI treatment outcomes. We identify two ICI-resistant spatiotypes in AYA patients with melanoma showing stroma-infiltrating lymphocytes (SILs) that are distinct from the adult TME. The SIL^high subtype was enriched in regulatory T cells in the peritumoral space and showed upregulated expression of immune checkpoint molecules, while the SIL^low subtype showed a lack of immune activation. We establish a young immunosuppressive melanoma score that can predict ICI responsiveness in AYA patients and propose personalized therapeutic strategies for the ICI-resistant subgroups. These findings highlight the distinct immunogenomic profile of AYA patients, and individualized TME features in ICI-resistant AYA melanoma that require patient-specific treatment strategies.

The incidence of melanoma in adolescents and young adults (AYAs) is disproportionately high, especially in western countries where it is the most frequent cancer diagnosis in those under 40 years of age[1,2]. The advent of immune checkpoint inhibitor (ICI) therapies has brought substantial improvements in patient outcome, where the overall survival (OS) for metastatic melanoma patients has improved from 10% to 50% with anti-PD-1-based immunotherapies when compared with traditional treatments[3–7]. However, the benefits of ICI therapies are highly variable and there are currently no reliable biological or clinical biomarkers that can predict treatment response. While various factors

such as previous therapy, functional status, tumor burden, and individual contraindications have prognostic implications and can potentially guide treatment decisions[8], the role of patient age in treatment efficacy remains unclear.

Studies of the effect of age on treatment outcomes have produced inconsistent results, and the majority of these studies have examined age cut-offs more relevant to older patients (e.g., 65–80 years)[9]. The Keynote 006[5] and CheckMate 067[10] trials demonstrated shorter progression-free survival (PFS) for patients aged <65 years who were treated with pembrolizumab or nivolumab monotherapy.

[1]Melanoma Institute Australia, The University of Sydney, Sydney, NSW, Australia. [2]Faculty of Medicine and Health, The University of Sydney, Sydney, NSW, Australia. [3]Charles Perkins Centre, The University of Sydney, Sydney, NSW, Australia. [4]Royal Prince Alfred Hospital, Sydney, NSW, Australia. [5]NSW Health Pathology, Sydney, NSW, Australia. [6]Centenary Institute, The University of Sydney, Sydney, NSW, Australia. [7]Westmead and Blacktown Hospitals, Sydney, NSW, Australia. [8]Royal North Shore Hospital, Sydney, NSW, Australia. [9]Mater Hospital, North Sydney, NSW, Australia. [10]These authors contributed equally: Georgina V. Long, Richard A. Scolyer, James S. Wilmott, Camelia Quek. ✉e-mail: camelia.quek@melanoma.org.au

Furthermore, a meta-analysis of 538 pembrolizumab monotherapy-treated metastatic melanoma patients identified response rates of 50% in patients aged <62 years compared to 63% in those ≥62 years old[11]. The same study observed higher immunosuppressive regulatory:cytotoxic T cell ratios in patients <60 years old, with regulatory T cell (T$_{reg}$) depletion leading to increased response to anti-PD-1 in younger mice. More recently, a retrospective study compared young adult melanoma patients ≤40 years to older age groups, and found no overall difference in immunotherapy response based on age, but young adults had a significantly higher response rate (53% vs 38%) and improved PFS (median 13.7 vs 4.0 months) with combination ICI (anti-PD-1+anti-CTLA-4) compared to monotherapy[12]. The same study also analyzed patients ≤30 years, which also showed better or trends towards better outcomes with combination ICI vs anti-PD-1 monotherapy (objective response rates of 57% vs 44%, PFS 13.8 vs 4.0 months, OS 82.1 vs 24.1 months)[12].

The limited studies on the use of ICIs in young melanoma patients suggest differences in response between older and younger patients and across regimens, but more comprehensive analyses of the tumor microenvironment (TME) at the multi-omic level are required to fully understand the biological basis of ICI response and resistance in AYAs. Advances have been made in understanding the TME features associated with ICI response in adult patients[13–15]. Biological factors such as greater degrees of ultraviolet light damage, and higher tumor mutation burden in older patients are predictors for better ICI efficacy[11,16]. It remains unclear to what extent the findings derived from adult melanoma patients are translatable to AYA patients.

Melanoma response to ICIs is influenced by the complex network of cells and signaling pathways within the TME[17]. AYA melanomas have shown higher frequencies of *BRAF* and *PTEN* mutations than older adult patients[18], which were associated with both T cell infiltration and immunosuppression in the TME[19–21]. In older cancer patients, *PTEN* mutations were correlated with a less favorable TME and reduced response to targeted BRAF/MEK inhibitors and ICI[22,23]. It remains unknown whether the immunological effects of these intrinsic mutations are associated with ICI outcome in AYA melanoma patients.

Extrinsic factors, such as the presence of immunosuppressive T$_{regs}$ can inhibit effector cell expansion and proinflammatory cytokine signaling[24–26]. Identification of immunosuppressive mediators within ICI-resistant AYA tumors may provide therapeutic strategies specific to young adult patients.

We conducted the most comprehensive study to date characterizing the immunogenomic profiles and spatial architecture of AYA (aged ≤30 years) melanoma, and comparing the AYA TME with older adult counterparts. Spatial and transcriptomic hallmarks associated with ICI resistance in younger patients were identified to facilitate the optimization of therapeutic strategies in order to overcome immunotherapeutic resistance. We identified the genomic, transcriptomic, and spatial cellular features of two major subtypes of ICI-resistant AYA tumors (stroma infiltrating lymphocyte [SIL] high and low tumor subtypes) and established an immunosuppressive score for AYA melanoma patients which differentiated the ICI response in this cohort. Additionally, we identified potential immune and non-immune based therapeutic targets as a proof-of-principle analysis for patient-specific treatment selection in AYA melanoma patients resistant to standard-of-care ICIs.

## Results
### Clinical characteristics and response to ICI
This study analyzed 47 AYA and 71 older adult patients with metastatic melanoma. The AYA group includes 28 patients who received ICI therapy in either the adjuvant or advanced setting (cohort 1) and 19 patients sourced from TGCA database with no accessible clinical record of immunotherapy treatment (cohort 3), and 71 adult patients received ICI in the advanced setting (cohort 2; Fig. 1a, Supplementary Data 1–2). The median age at diagnosis was 26 years (interquartile range [IQR], 22–28) for the AYA cohort and 57 years (IQR, 46–65) for the adult cohort. Gender distribution was similar between the two cohorts, with more males in both groups (62% in the AYA cohort and 66% in the adult cohort, Fisher's exact test, $P = 0.70$). The clinically characterized tumor molecular subtypes included *BRAF*, *NRAS*, *KRAS*, *KIT*, and *PIK3CA* mutant melanomas, with *BRAF* mutant melanomas

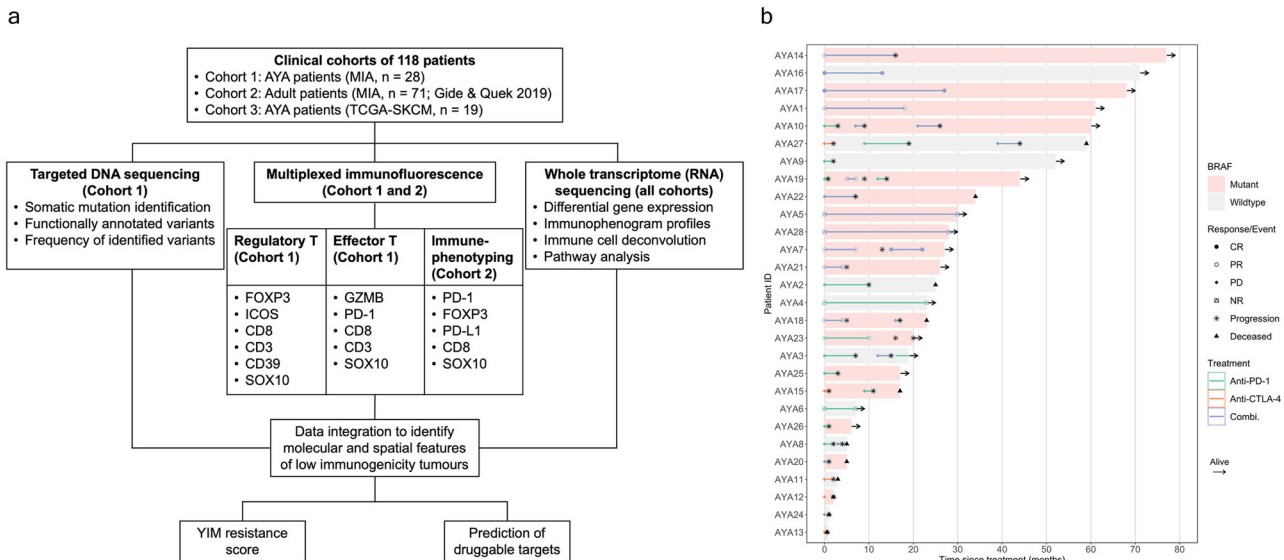

**Fig. 1 | Study outline and immunotherapy treatment response in AYA melanoma patients. a** A flow diagram of the study design, formalin-fixed paraffin-embedded (FFPE) melanoma samples were collected at pre-treatment baseline for MIA cohorts 1 and 2, and cohort 3 data was retrieved from The Cancer Genome Atlas Skin Cutaneous Melanoma (TCGA-SKCM) project. The samples underwent sequencing and immunofluorescence staining, before integrated data analysis as outlined. **b** Swimmer plot of AYA patients from MIA (cohort 1) showing the patients'

clinical *BRAF* mutation status and immunotherapy treatments (combination anti-PD-1 and anti-CTLA-4, combi). For patients treated in the advanced stage disease setting, immunotherapy RECIST response (complete response, CR; partial response, PR; progressive disease, PD) is shown, for patients treated with adjuvant immunotherapy (AYA 4, 5, 6, 22, 25, 26, 28), disease recurrence is labeled as PD and no recurrence is labeled as NR.

**Table 1 | Clinicopathologic characteristics of AYA and adult patients treated with ICI in the advanced setting**

| Patient characteristics | ICI advanced setting | | P value (two-sided Fisher's exact test) |
|---|---|---|---|
| | AYA (n = 21) | Adult (n = 71) | |
| *BRAF* | | | |
| Wildtype | 8 (38%) | 35 (49%) | |
| Mutant | 13 (62%) | 24 (34%) | 0.1273 |
| NR | 0 | 12 (17%) | |
| Baseline LDH | | | |
| Normal | 13 (62%) | 47 (66%) | |
| Elevated | 7 (33%) | 24 (34%) | >0.9999 |
| NR | 1 (5%) | 0 | |
| AJCC pathologic stage | | | |
| M0, M1a, M1b | 8 (38%) | 33 (46%) | |
| M1c, M1d | 13(62%) | 38 (54%) | 0.6192 |
| Treatment | | | |
| Anti-PD-1 | 7 (33%) | 34 (48%) | 0.0001 |
| Anti-CTLA-4 | 5 (24%) | 0 | |
| Combination therapy[a] | 9 (43%) | 37 (52%) | |
| Best RECIST response | | | |
| CR | 2 (10%) | 18 (25%) | 0.0474 |
| PR | 6 (29%) | 23 (32%) | |
| SD | 0 | 10 (14%) | |
| PD | 13 (62%) | 20 (28%) | |
| Progressed | | | |
| Yes | 18 (86%) | 43 (61%) | 0.0942 |
| No | 3 (14%) | 20 (28%) | |
| NR | 0 | 8 (11%) | |

AJCC metastasis staging: M0 = no evidence of distant metastasis; M1a = distant metastasis to skin and soft tissue, M1b = distant metastasis to lung, M1c = distant metastasis to non-central nervous system (CNS) visceral sites. M1d = distant metastasis to CNS.
*NR* not reported, *LDH* lactate dehydrogenase.
[a]Anti-PD-1+anti-CTLA-4.

**Table 2 | Clinicopathologic characteristics of all AYA and adult patients treated with ICI**

| Patient characteristics | AYA (n = 27) | | Adult (n = 71) | |
|---|---|---|---|---|
| | ICI non-resistant (n = 11) | ICI resistant (n = 16) | ICI non-resistant (n = 45) | ICI-resistant (n = 26) |
| Age (median, IQR) | | | | |
| At diagnosis | 22, 21–27 | 26, 24–29 | 58, 48–66 | 51, 44–62 |
| At start of ICI treatment | 27, 23–30 | 30, 26–34 | 66, 52–73 | 55, 50–67 |
| Gender | | | | |
| Female | 2 (18%) | 8 (50%) | 16 (36%) | 8 (31%) |
| Male | 9 (82%) | 8 (50%) | 29 (64%) | 18 (69%) |
| *BRAF* | | | | |
| Wildtype | 2 (18%) | 6 (38%) | 24 (53%) | 11 (42%) |
| Mutant | 9 (82%) | 10 (63%) | 13 (29%) | 11 (42%) |
| NR | 0 | 0 | 8 (18%) | 4 (16%) |
| Baseline LDH | | | | |
| Normal | 9 (82%) | 9 (56%) | 32 (71%) | 15 (58%) |
| Elevated | 2 (18%) | 5 (31%) | 13 (29%) | 11 (42%) |
| NR | 0 | 2 (13%) | 0 | 0 |
| AJCC pathologic stage | | | | |
| M0, M1a, M1b | 7 (64%) | 7 (44%) | 25 (56%) | 8 (31%) |
| M1c, M1d | 4 (36%) | 9 (56%) | 20 (44%) | 18 (69%) |
| Treatment | | | | |
| Anti-PD-1 | 2 (18%) | 8 (50%) | 20 (44%) | 14 (54%) |
| Anti-CTLA-4 | 0 | 5 (31%) | 0 | 0 |
| Combination therapy[a] | 9 (82%) | 3 (19%) | 25 (56%) | 12 (46%) |
| Best RECIST response[b] | | | | |
| CR | 2 (18%) | 0 | 18 (40%) | 0 |
| PR | 6 (55%) | 0 | 23 (51%) | 0 |
| SD | 0 | 0 | 4 (9%) | 6 (23%) |
| PD | 0 | 13 (81%) | 0 | 20 (77%) |
| Progressed | | | | |
| Yes | 5 (45%) | 16 (100%) | 17 (38%) | 26 (100%) |
| No | 6 (55%) | 0 | 20 (44%) | 0 |
| NR | 0 | 0 | 8 (18%) | 0 |

*IQR* interquartile range, *NR* not reported, *LDH* lactate dehydrogenase.
[a]Anti-PD-1+anti-CTLA-4
[b]For patients treated in the advanced setting only.

being more prevalent in AYA compared to adult patients (77% vs 41%; Fisher's exact test, $P = 0.0003$). AYA patients were diagnosed at a lower metastasis (M) stage compared to adult patients (Fisher's exact test, $P = 0.037$).

In the advanced treatment setting, ICI response was defined as RECIST complete response (CR), partial response (PR), or stable disease lasting more than 6 months prior to progression, as previously described[15]. The distribution of *BRAF* mutation and baseline lactate dehydrogenase levels were similar between responders and non-responders in AYA and adult groups (Table 1). Overall, objective response to ICIs was lower among 21 AYA patients compared to 71 adult patients (38% vs 63%; Fisher's exact test, $P = 0.047$; Table 1). However, for AYA patients, combination ICI therapy had a significantly higher response rate of 78% (7/9) compared to 5% (1/12) with monotherapy treatment (Fisher's exact test, $P = 0.022$). In comparison, the response rate for combination ICI versus monotherapy in adult patients was 68% (25/37) versus 59% (20/34) (Fisher's exact test, $P = 0.47$). There was a trend of shorter PFS in AYA patients compared to adults (median PFS = 3.4 months vs 10 months, log-rank $P = 0.059$), OS was similar between the two groups (Supplementary Fig. 1a, b). Of the 8 AYA patients who responded to ICI, only 3 (14%) did not experience subsequent disease progression (2 CR and 1 PR), all of whom received combination ICI (anti-PD-1+anti-CTLA-4) (Supplementary Data 1, Fig. 1b).

AYA cohort 1 included patients treated in the adjuvant setting for downstream analysis (Methods). Patients were considered responders (ICI non-resistant) according to the RECIST criteria above (for advanced setting), in the adjuvant setting, patients with a recurrence-free survival (since treatment initiation) of more than 12 months were considered "non-resistant" (Table 2, Supplementary Data 1), in line with the prior study[27]. Out of all the AYA patients with a reported clinical outcome to immunotherapy either in the adjuvant or advanced setting ($n = 27$), 11 (41%) were ICI non-resistant and 16 (59%) were ICI-resistant. In comparison, the proportion of non-resistant patients was higher in the adult cohort (63%; $\chi^2$ test, $P < 0.0001$). Non-resistant patients were more often treated with the combination ICI than monotherapy in both AYA (81.8% vs 18.8%) and adult (56% vs 46.2%) cohorts, with this difference more pronounced in AYA patients ($\chi^2$, $P = 0.0095$).

Overall, survival outcome was poor among AYA patients treated with ICI in the advanced setting (Fig. 1b). AYA patients had a median PFS of 5.6 months, and a median OS of 60.7 months. Among the progressors, 8 patients received subsequent lines of ICI therapy treatment and none of them responded, indicating a high rate of persistent immunotherapy resistance and progressive disease. No survival difference was observed between *BRAF* wildtype and mutant patients

(Supplementary Fig. 1c, d); consistent with the clinical trials[4], OS was significantly longer among patients treated with anti-PD-1 or combination ICI compared with anti-CTLA-4 monotherapy ($P = 0.0028$), while PFS was longer in patients treated with combination ICI relative to anti-PD-1 and anti-CTLA-4 monotherapy ($P = 0.0014$) (Supplementary Fig. 1e, f). Together, the response and survival data suggest a possible benefit of combination ICI treatment for AYA melanoma patients.

## Enrichment of FOXP3$^+$ T cells within the AYA melanomas compared to adults

To assess the immune landscape of the AYA and adult melanomas at baseline, we included transcriptomic data from a further cohort of AYA patients (cohort 3, $n = 19$) sourced from the TCGA database. We used the transcriptomic-based CIBERSORT immune cell deconvolution to compare the estimated immune cell proportions in AYA ($n = 47$) vs adults ($n = 71$) (Fig. 2a, b; Supplementary Data 3–4). While the melanomas from AYA and adult patients displayed similar proportions of CD8 T cells (mean relative percentage = 3.62% vs 4.61%, $P = 0.30$), AYA melanomas contained higher proportions of $T_{reg}$ cells (Fig. 2c; mean relative percentage = 2.62% vs 0.893%, $P = 0.0015$). Additionally, AYA melanomas had higher proportions of naïve B cells (mean relative percentage = 3.21% vs 1.24%, $P < 0.0001$), and lower proportions of plasma cells (mean relative percentage = 0.766% vs 4.36%, $P < 0.0001$) and M2 macrophages (mean relative percentage = 9.64% vs 15.5%, $P < 0.0001$) compared to adults (Supplementary Fig. 2a).

To further evaluate the T cell composition and spatial location in the TME, we performed multiplex immunofluorescence (mIF) staining (Fig. 2d, e; Supplementary Fig. 2b) on the same baseline tumor resections that were used for sequencing (Fig. 1a, cohort 1 and 2), and found a significant enrichment of FOXP3$^+$ $T_{regs}$ in AYA compared to adult melanomas (Fig. 2f; median = 110 vs 12.3 cells/mm$^2$, $P = 0.023$), and a trend of lower CD8$^+$:FOXP3$^+$ T cell ratio in AYA melanomas (Fig. 2g). Comparison between the response groups showed that both ICI-resistant and non-resistant AYA melanomas harbored higher densities of FOXP3$^+$ $T_{regs}$ compared to resistant (median = 72.8 and 122 vs 9.38 cells/mm$^2$, $P = 0.0006$ and 0.001) and non-resistant (median = 15.1 cells/mm$^2$, $P = 0.0048$ and 0.0069) adult melanomas (Supplementary Fig. 2c). There was a trend of higher CD8$^+$:FOXP3$^+$ T cell ratio in ICI non-resistant compared to resistant adult patients (median = 2.18 vs 5.96, $P = 0.058$), possibly affected by the small cohort size, this trend was not statistically significant in the AYA cohort (median = 2.74 vs 5.10, $P = 0.77$). The CD8$^+$:FOXP3$^+$ T cell ratio in ICI-non-resistant AYA melanomas was similar to the adult groups, while the resistant AYA group showed a trend of lower CD8$^+$:FOXP3$^+$ cell ratio compared to adult non-resistant melanomas ($P = 0.085$) (Supplementary Fig. 2d). To determine whether $T_{reg}$ cell density was different across age groups, we stratified the baseline AYA and adult patients into four age groups (15–30, 31–45, 46–60, 61–84), this showed that the $T_{reg}$ enrichment was only evident in the 15–30 (AYA) age group (Fig. 2h), with no difference across older adult age groups ($P = 0.33$).

We also correlated the transcriptomic $T_{reg}$ proportion estimates with the mIF $T_{reg}$ quantification to determine the concordance of the approaches. In AYA melanomas, the expression of key gene markers for $T_{regs}$ (*CD3D*, *FOXP3*) strongly correlated with CD3$^+$FOXP3$^+$ cell density (Supplementary Fig. 2e; Spearman's $\rho = 0.74$, $P < 0.0001$). Furthermore, the expression of functional $T_{reg}$ markers (*CD3D*, *FOXP3*, *ICOS*) also correlated with its mIF (CD3$^+$FOX3$^+$ICOS$^+$ cell) density (Supplementary Fig. 2f; Spearman $\rho = 0.68$, $P = 0.00011$). The high proportion of $T_{reg}$ cells demonstrated the distinct TME profile of AYA melanomas.

## $T_{regs}$ are localized predominantly at the peritumoral margin of ICI-resistant AYA melanomas

We observed that the majority of $T_{regs}$ (CD3$^+$FOXP3$^+$) in ICI-resistant AYA melanomas were localized in the stromal side of the tumor invasive margin (peritumor region; Fig. 3a), as shown by the significantly higher peritumoral $T_{reg}$ cell density compared to the intratumoral (Fig. 3b; median = 99.4 vs 17.3 cells/mm$^2$, $P = 0.0008$). There was a non-significant, but numerically higher peritumoral $T_{reg}$ density in the AYA ICI-resistant patients compared to the AYA non-resistant patients (Fig. 3c; median = 68 vs 102 cells/mm$^2$, $P = 0.37$). $T_{reg}$ density was not different between response groups in the intratumor region (Supplementary Fig. 3a).

Since $T_{regs}$ can impair effector T-cell function[11], we then correlated the peritumoral $T_{reg}$ densities with a transcriptional score for "Tumor Immune Dysfunction and Exclusion"[28]. The AYA T cell dysfunction score was most strongly correlated with peritumoral $T_{reg}$ densities (Fig. 3d; Spearman $\rho = 0.54$, $P = 0.003$), and was also correlated with the total and intratumoral $T_{reg}$ densities (Supplementary Fig. 3b, c).

## Subtypes of immunologically suppressed ICI-resistant AYA tumors show distinct T cell infiltration

Given the pressing need to improve the therapeutic options for AYA melanoma patients who fail the standard-of-care immunotherapies, we sought to characterize the TME features of resistant patients. The transcriptomic-based immunophenogram score[29] was used to evaluate the overall immunogenicity of ICI AYA melanomas (Supplementary Fig. 4), which qualitatively stratified the innately resistant tumors into high and low immunogenicity groups (Fig. 3e, f). Principal component analysis of transcriptomic profiling identified two subgroups of AYA ICI-resistant tumors (Fig. 3g). Group 1 melanomas were characterized by higher expression of effector cell, suppressor cell and checkpoint genes, which reflected an overall similar profile compared to the non-resistant patients (Fig. 3e, Supplementary Fig. 4a, b); while Group 2 TME was characterized by lower transcriptomic expression of checkpoint molecules and immune cell markers (Fig. 3f, Supplementary Fig. 4c). All Group 2 patients showed the downregulation of *PD1* and at least one other co-inhibitory immune checkpoint, such as *CTLA4*, lymphocyte activation gene 3 (*LAG3*), T cell immunoreceptor with Ig and ITIM domains (*TIGIT*), T cell immunoglobulin and mucin protein 3 (*TIM3*) and Indoleamine 2,3 dioxygenase (*IDO1*) compared to Group 1 (Supplementary Fig. 4b, c).

Multiplex IF was used to compare the spatial (peritumoral and intratumoral) densities of T cells in ICI non-resistant and resistant subgroups of AYA melanomas (Supplementary Fig. 5a, b, Supplementary Data 5), identifying high stroma infiltrating lymphocytes (SILs) in Group 1 patients. Intratumorally, the non-resistant and SIL$^{high}$ group (Group 1) had significantly higher densities of $T_{reg}$ populations compared to the SIL$^{low}$ group (Supplementary Fig. 5a), including CD3$^+$FOXP3$^+$ (median = 21.2 and 95.4 vs 4.71 cells/mm$^2$, $P = 0.027$ and 0.0017), CD3$^+$CD8$^-$FOXP3$^+$ (median = 18.2 and 68.6 vs 4.65 cells/mm$^2$, $P = 0.027$ and 0.0017) and CD3$^+$CD8$^-$FOXP3$^+$ICOS$^+$ (median = 11.7 and 53.2 vs 1.48 cells/mm$^2$, $P = 0.015$ and 0.0047). Overall, there was a trend of higher infiltration of $T_{reg}$ populations in SIL$^{high}$ group compared to non-resistant group, suggestive of tumor recruitment of immunosuppressive cells.

Peritumorally, the non-resistant group and SIL$^{high}$ group had significantly higher densities of multiple $T_{reg}$ and tumor-specific CD39$^+$ T cell[30] populations compared to SIL$^{low}$ group (Supplementary Fig. 5b), including CD3$^+$FOXP3$^+$ (median = 109 and 257 vs 22.0 cells/mm$^2$, $P = 0.048$ and 0.0010), CD3$^+$CD8$^-$FOXP3$^+$ (median = 93.4 and 233 vs 21.4 cells/mm$^2$, $P = 0.048$ and 0.0017) CD3$^+$CD8$^-$FOXP3$^+$ICOS$^+$ (median = 61.3 and 126.2 vs 6.29 cells/mm$^2$, $P = 0.015$ and 0.00025), CD3$^+$CD39$^+$ (median = 417 and 864 vs 15.4 cells/mm$^2$, $P = 0.0070$ and 0.0047) and CD3$^+$CD8$^-$CD39 (median = 94.0 and 70.5 vs 1.28 cells/mm$^2$, $P = 0.0048$ and 0.0047). Overall, SIL$^{low}$ group had low peritumoral densities of all T cell subtypes, corresponding to their low immunogenic state.

The SIL$^{high}$ ICI-resistant melanomas also had similar densities of intratumoral CD3$^+$CD8$^+$ cytotoxic T cells and higher densities of

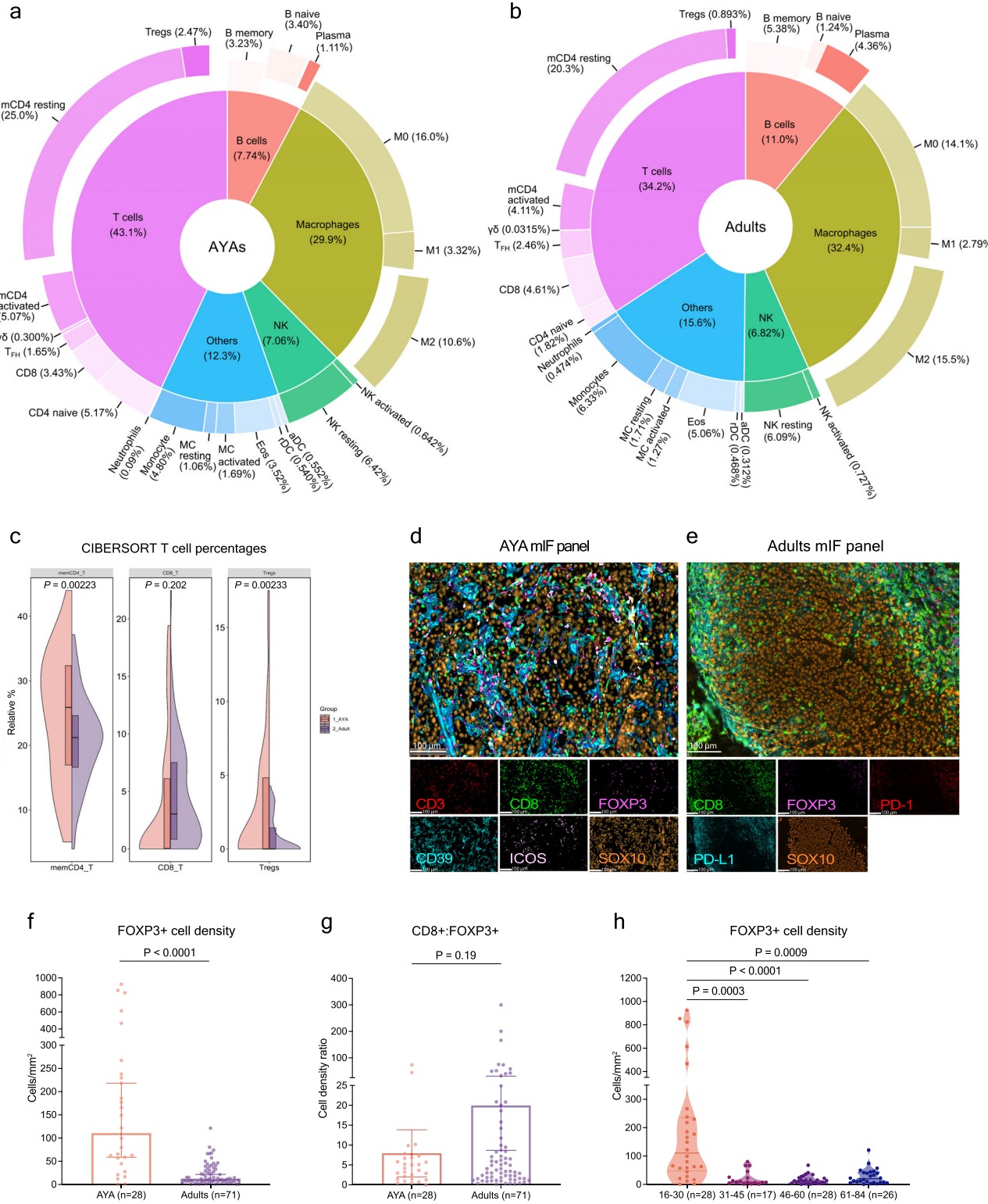

**Fig. 2 | AYA melanomas harbor distinct proportions of T cell and innate immune cell infiltrates compared to older adult melanomas. a, b** CIBERSORT immune cell deconvolution analysis of whole transcriptome sequencing data. Pie chart displays the relative percentages of 22 immune cell subtypes in AYA ($n = 47$) (**a**) and adult ($n = 71$) (**b**) melanoma samples at treatment-naïve baseline, main categories are ordered by cell type and sub-categories by relative abundance. **c** Violin graph with boxplot showing the median and interquartile range of the proportion of key T cell subtypes between AYA ($n = 47$) and adult ($n = 71$) melanoma patients, $P$ values are derived from multiple unpaired $t$ test. **d**, **e** Multiplex immunofluorescence (mIF) analysis of AYA and adult tumors. Representative mIF images of AYA (**d**) and adult (**e**) melanoma samples stained with respective panels (Fig. 1a); scale bars are 100 μm. mIF quantitative analysis, Mann–Whitney test comparing the FOXP3⁺ cell densities (cells/mm²) (**f**), Unpaired $t$ test comparing the ratio of CD8:FOXP3 expressing cells between AYA and adult patients (**g**), and Dunn's multiple comparisons test comparing FOXP3⁺ cells densities across age groups (**h**); bar plot shows median and 95% CI, violin graph shows median and interquartile range.

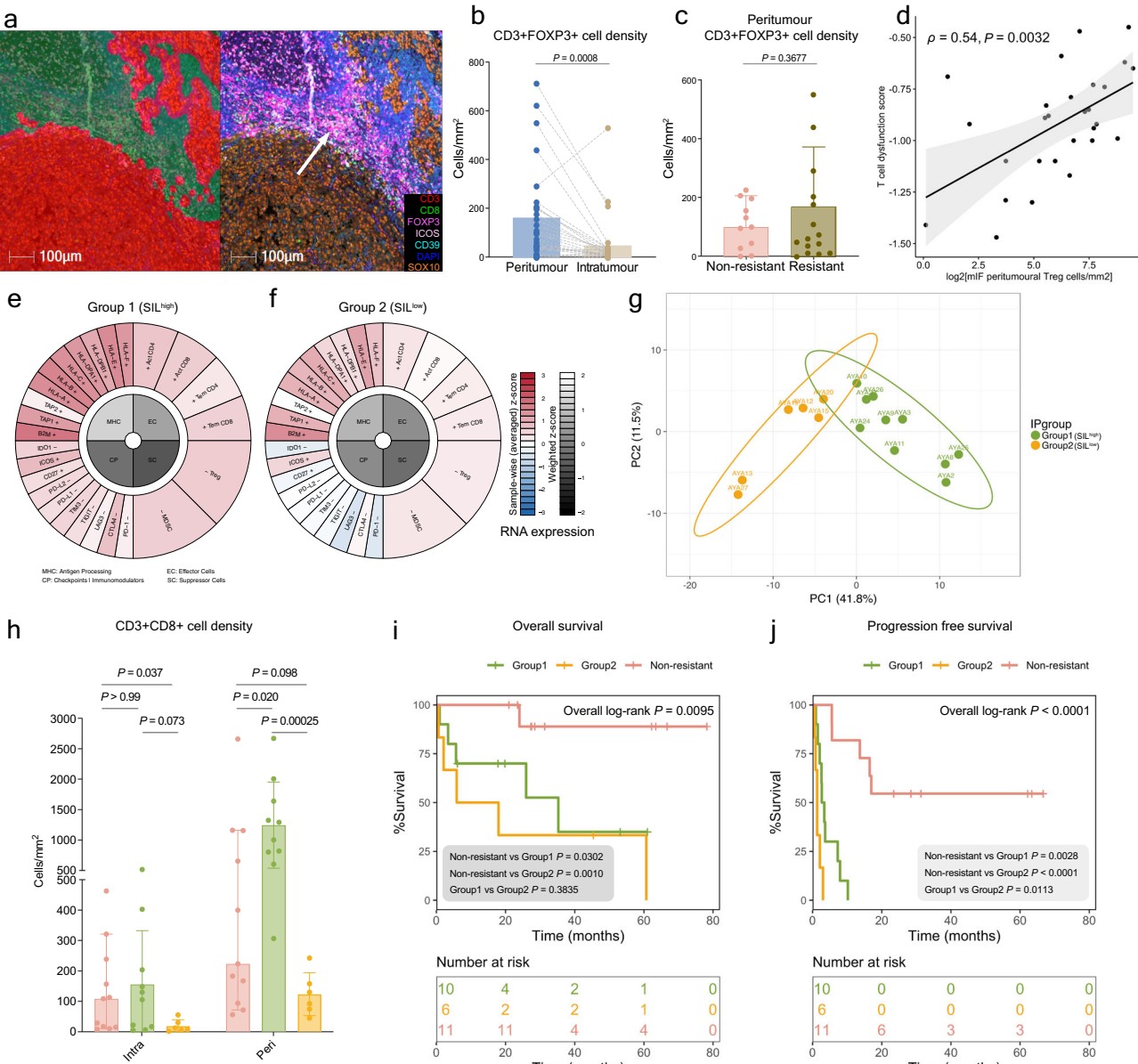

**Fig. 3 | Resistant hotspot of peritumoral T$_{regs}$ and immunosuppressed spatiotypes of AYA melanoma at pre-treatment baseline. a** Representative mIF image of the enrichment of T$_{regs}$ (white arrow) in the peritumor region from an AYA ICI-resistant patient (AYA8); red and green masking (left) represents regions classified as tumor and peritumor respectively. **b** Mann–Whitney test comparing T$_{reg}$ cell densities in the peritumor and intratumor regions in AYA melanoma ($n = 28$). **c** Unpaired t test comparing the peritumor T$_{reg}$ cell densities in AYA ICI-non-resistant ($n = 11$) and ICI-resistant ($n = 16$) patients; bar plot shows median and 95% CI. **d** Two-sided spearman's correlation of TIDE T cell dysfunction scores and peritumor T$_{reg}$ cell densities in AYA patients ($n = 28$). Error band shows 95% CI. Immunophenogram gene expression of AYA ICI-resistant patients; representative

immune expression profiles of Group1 ($n = 10$) (**e**) and Group 2 ($n = 6$) (**f**) patients; Immunophenograms for each patient are shown in Supplementary Fig. 4. **g** Principal component analysis of Group 1 and Group 2 ICI-resistant patients based on immunophenogram grouping (IP group). **h** Multiple Mann–Whitney test comparing the intratumoral (intra) and peritumoral (peri) cytotoxic T cell densities between AYA ICI non-resistant patients ($n = 11$), and Group 1 ($n = 10$) and Group 2 ($n = 6$) resistant patients; granzyme B, GZMB; bar plot shows median and 95% CI. Kaplan–Meier curves comparing the overall (**i**) and progression-free (**j**) survival between AYA ICI non-resistant patients ($n = 11$), and Group 1 ($n = 10$) and Group 2 ($n = 6$) resistant patients.

peritumoral CD3$^+$CD8$^+$ T cells (median = 1250 vs 623 cells/mm$^2$, $P = 0.20$) and CD3$^+$PD1$^+$ T cells (median = 508 vs 88.4 cells/mm$^2$, $P = 0.036$) compared to the ICI non-resistant group (Fig. 3h, Supplementary Fig. 5a, b), indicating potential antitumor immune response that was hindered by suppressive T$_{reg}$ enrichment. Since M2 macrophages and plasma cells were identified by the CIBERSORT analysis as significantly decreased in AYA melanoma compared to adults, and naïve B cells significantly increased, we also compared the estimated cell proportions between ICI non-resistant and resistant groups of AYA patients (Supplementary Fig. 5c). Of note, non-resistant patients had

higher M2 macrophage proportions compared to the SIL$^{high}$ group (median relative percentage = 12.3% vs 3.17%, $P = 0.0021$), and higher plasma cell proportions compared to both SIL$^{high}$ and SIL$^{low}$ groups (median relative percentage = 2.42% vs 0% and 0%, $P = 0.013$ and 0.012). The distinct cell proportions could be indicative of immune regulation in the TME, but requires further experimental validation to confirm the phenotype and spatial location of these cells.

Lastly, we compared the survival outcomes between ICI non-resistant, resistant, and SIL subgroups of AYA patients. The SIL$^{high}$ resistant group had similar OS but longer PFS ($P = 0.011$) compared to

the SIL$^{low}$ resistant group (Fig. 3i, j). We also stratified the ICI non-resistant patients by SIL category, given the small cohort, SIL spatiotype did not differentiate survival outcome in ICI non-resistant patients. However, the survival analysis showed significantly longer OS and PFS in the non-resistant SIL$^{high}$ group compared to the resistant groups, as well as a longer PFS in the non-resistant SIL$^{low}$ group compared to the resistant SIL$^{low}$ group (Supplementary Fig. 5d, e). When AYA patients were stratified by SIL spatiotype regardless of ICI outcome, the SIL$^{high}$ group showed a trend of longer OS and a near-significant longer PFS ($P = 0.0504$) compared to SIL$^{low}$ group (Supplementary Fig. 5f, g). The predictive and prognostic powers of SIL spatiotype warrant further investigation in bigger cohorts.

### Intrinsic and extrinsic hallmarks of resistance in AYA patients who progressed on ICI immunotherapies

Analysis of the somatic mutation profile of AYA patients (Supplementary Data 11) revealed potential intrinsic factors associated with immunotherapy response and resistance, including high rates of *BRAF* (62%) and *PTEN* (21%) mutations (Supplementary Data 12), which are known to influence the efficacy of ICI therapy[20,21,31]. However, we did not observe any associations between the most frequent somatic mutations (*BRAF, CDKN2A, PTEN, NRAS*) and T$_{reg}$ infiltration (Supplementary Fig. 6a) in AYA melanoma. This was in concordance with our finding of no association between *BRAF* mutation status and patient survival (Supplementary Fig. 1c, d), and no distinct mutational differences were observed between patients in the two ICI-resistant subtypes (Fig. 4a). Similar to findings in adults[32], AYA melanomas harbored frequent mutations in the *TERT* promoter region (Supplementary Fig. 8), which showed a trend of co-occurrence with *BRAF* mutations (Fisher's exact test, $P = 0.066$).

We then queried the extrinsic microenvironmental features of immunosuppression via differential expression analysis of the SIL$^{high}$ and SIL$^{low}$ AYA ICI-resistant patients (Fig. 4b, Supplementary Data 6). In accordance with their more inflamed immunophenotype and enrichment of peritumoral T$_{regs}$, the SIL$^{high}$ AYA patients showed the upregulation of immunosuppressive anti-inflammatory cytokines and receptors (*TGFB1, TGFB2, TGFBR2, IL10RA*). In contrast, SIL$^{low}$ AYA patients showed transcriptomic features of poor immunogenicity via the downregulation of T cell receptor signaling (*ZAP70, CD247, CD3, CD8, PDCD1, CARD1, IFNG, TNF*), cytokine-cytokine receptor interaction (*IL10RA, CXCL9, IL12R, IL-7, IL-26, IFNLR1, FASLG, IL4R*) and cell adhesion molecules (*VCAM, HLA, CD6, SELL, CADM3*) among other immune signaling pathways (Fig. 4b, c, Supplementary Fig. 6b, c, Supplementary Data 7). Gene Ontology analysis also showed the downregulation of immunological processes including lymphocyte activation, immune response, cell-cell adhesion and the suppression of molecular signaling receptor activity (Supplementary Fig. 6b, Supplementary Data 7) in SIL$^{low}$ melanomas. Collectively, we demonstrate the major extrinsic activities that contribute to the state of immunosuppression in AYA ICI-resistant patients—altered signaling potentially mediated by T$_{regs}$ dampened the inflammatory response in the SIL$^{high}$ subgroup, while the lack of immune activation is dominant in the SIL$^{low}$ subgroup.

### ICI-resistant AYA patients with high young immunosuppressive melanoma (YIM) scores express alternative immune and non-immune drug targets

Given the distinct TME landscape of AYA melanomas, we established a gene expression-based scoring strategy to stratify AYA patients into ICI resistant and non-resistant groups. Given the limited cohort, we used a rank-based gene scoring approach that is independent of sample size[33]. The selected genes used for scoring were identified by differential expression analysis between patients who progressed on ICI therapy versus CR patients who did not progress (no primary or acquired ICI resistance). Genes that were highly expressed in PD (primary resistance) patients were positively weighted in the scoring

framework, and highly expressed genes in CR patients were negatively weighted in the calculation of the YIM score (Supplementary Data 8–9).

We then compared the predictive value of the YIM score to previously reported gene signatures of innate anti-PD1 resistant (IPRES), and the combined immune gene signature (IMMU) derived by integrating the chemokine, interferon-γ, T effector, T cell inflamed, B-catenin, TGFβ and immunosuppression gene signatures (Supplementary Fig. 6d, e, Supplementary Data 8). The YIM score demonstrated superior predictive value for ICI response (CR and PR vs PD) in AYA patients compared to the IPRES and IMMU scores (AUC: 79% vs 63% and 45% respectively) (Supplementary Fig. 6e). Lower YIM score further showed a trend of longer disease specific survival in TCGA AYA (cohort 3) patients (Supplementary Fig. 6f). With less than 1% of genes overlapping with previously reported gene signatures, the YIM score represents the unique biology of AYA melanoma patients in orchestrating microenvironmental ICI resistance (Supplementary Fig. 7a, b, Supplementary Data 8).

To assess the expression of potential therapeutic targets for ICI-resistant AYA patients, we performed drug gene interaction prediction using the transcriptomic and mutational profiles, which identified 27 actionable targets with currently approved oncolytic drugs (Fig. 4d, Supplementary Data 12). Given the lack of therapeutic progress for AYA cancer patients in the past decades, our data here shows the potential room for improvement. All ICI-resistant AYA patients expressed at least one currently druggable target at baseline, and their SIL$^{high}$ versus SIL$^{low}$ status might indicate what type of follow-up treatment would be most successful after the patient fails ICI. In line with their state of increased immune activation, a proportion of SIL$^{high}$ patients expressed alternative ICI targets (including *LAG3, IDO, TIGIT*), and have the potential to benefit from bi-specific T-cell engagers (BiTEs); SIL$^{low}$ tumors with overall dampened immunogenicity showed potential drug gene interactions for immune augmentation targets (*IL2RG, TNFRSF9*) and mutational targets (*BRAF, PTEN, NRAS, TP53*). More multi-omic profiling for AYA patients will facilitate the development of novel drugs and personalized treatment plans. As a general example, we summarized our TME profiling and drug interaction findings for ICI-resistant AYA melanoma patients into a treatment switch plan to exemplify future guides for translational research and clinical trial designs (Supplementary Fig. 6g).

## Discussion

ICI therapy has emerged as the standard of care for many advanced cancers, including melanoma, and has shown promising long-term responses in some patients. However, high rates of ICI resistance remains a significant obstacle, particularly for young melanoma patients[34]. In this study, we conducted an integrated analysis of molecular expression, spatial tumor-immune microenvironment, and clinical features to characterize ICI-resistant AYA patients. Our findings revealed two distinct groups of AYA melanoma: the SIL$^{high}$ group with high infiltration of peritumoral T$_{regs}$, and the SIL$^{low}$ group with a general lack of immune induction in the TME. Using the multi-omic approach, we were able to summarize the baseline immunosuppressive state of AYA melanomas into the YIM score, and perform the first proof of principle analysis towards ICI outcome prediction and personalized drug target discovery for AYA patients.

Although age and gender have been reported to affect the efficacy of ICI treatment[35], their impact on the immunosuppressive functions in the TME remains unclear. Reduced immunosurveillance and increased T$_{regs}$ have been observed in the skin of older individuals as an immunosenescent process[36], but age-dependent changes in T$_{reg}$ number and function at sites of chronic inflammation remains poorly understood. In line with previous study, we observed an increased proportion of FOXP3+ T$_{regs}$ in AYA patients compared to older adults, resulting in a reduced CD8:FOXP3 T cell ratio[11]. This increase in T$_{reg}$ proportions was

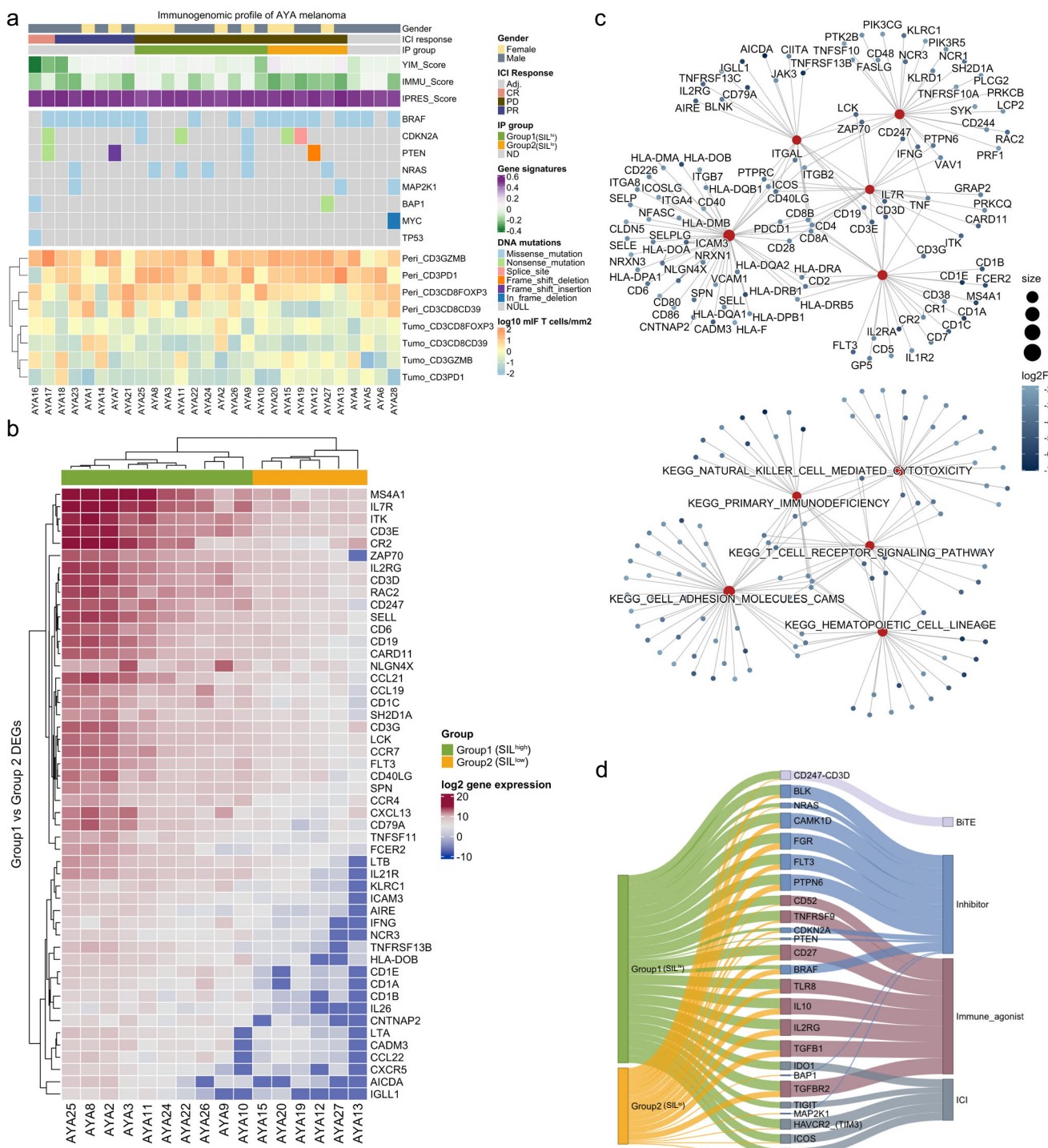

**Fig. 4 | Predicting immunotherapy resistance with the young immunosuppressed melanoma (YIM) score and drug target identification to overcome therapy resistance. a** Heatmaps showing the gender, immunotherapy response for non-adjuvant (Adj.) treatment (complete response CR, partial response PR, progressive disease PD), immune phenotype (IP) group, gene signature scores, somatic variants, and mIF T cell densities (peritumoral, peri; intratumoral, tumo) of AYA melanomas (*n* = 28); not determined, ND. **b** Heatmap of the top 50 differentially expressed genes between Group 1 and Group 2 ICI-resistant AYA patients which are correlated with KEGG pathways; top genes with adjusted-*P* < 0.05 (Wald test *P* values adjusted for multiple testing) were ranked by absolute fold change. **c** Top

pathways in KEGG gene set enrichment analysis (adjusted-*P* < 0.05). *P* values are calculated based on the hypergeometric distribution model and adjusted for multiple testing with the Benjamini Hochberg method. Dot size represents gene set size and color scale represents log2 fold change. **d** Sankey diagram highlighting the alternative drug-gene interactions in AYA ICI resistant patients (Supplementary Data 12); left represents the total number of druggable gene and mutational targets in SIL^high and SIL^low groups, middle shows targetable expressed or mutated markers, right shows the drug categories; line thickness represents the proportion within each group, for example, line will have a thickness of 1 if one patient expressed the drug target; bi-specific T cell engager, BiTE.

primarily observed in AYA patients aged 16–30 years old. These findings suggest that the presence of FOXP3 + T cell subtypes in some AYA tumors may impede the success of anti-PD-1-based therapies by suppressing CD8+ effector T cell function, thus AYA patients can potentially benefit from the use of combination ICI to overcome immunosupression[37].

Our study highlighted the unique TME in AYA melanomas, which may contribute to the differences in treatment outcomes to combination versus monotherapy ICI in this patient population. We identified an increase in $T_{regs}$ within the peritumoral region of ICI-resistant AYA melanomas, which may suppress effector T-cell function and prevent their intratumoral migration[24]. This is evident by the inflamed transcriptional phenotype, but with high peritumoral $T_{regs}$ and T cell dysfunction scores[38]. The upregulation of T cell receptor and cytokine receptor signaling pathways, including ICOS expression, further supports the immunosuppressive role of Tregs in the TME[26]. These results suggest the potential to reshape the immune TME with additional immune augmentation, such as combination or alternative ICI therapies. Similar to a previous study on young adult melanoma[12], we observed that AYA patients treated with combination ICI therapy had a higher response rate and longer PFS than monotherapy, which may be associated with $T_{reg}$ enrichment in the TME. These findings support the use of combination therapy as a potential strategy to overcome $T_{reg}$-mediated immunosuppression and improve treatment outcomes in AYA melanoma patients.

In contrast, immune signaling pathways were suppressed in ICI-resistant SIL^low tumors, which also lacked the expression of *IFNG*, *IFNLR1*, *TNF* and *TNFR* genes, limiting the pro-inflammatory immune stimulating activities within the TME, which is corroborated by our findings of poor lymphocyte infiltration and checkpoint expression in the SIL^low melanomas. Clinical decisions for treating SIL^low AYA melanomas with ICI standard therapies should be carefully considered, as their melanomas demonstrate the immune features associated with primary resistance[15]. Future clinical trials may consider leveraging SIL^low patients' genomic profiles at baseline to treat on a personalized basis. Increased antigen expression and TILs have also been observed post treatment with targeted therapies which may improve susceptibility to ICI when given in combination[22]. Further immune augmentation for SIL^low patients may be achieved with immune agonistic drugs or adoptive cell therapy[39].

Previous research on the tumor intrinsic disparities between cancers in AYA and adults have yielded mixed results. Some studies found an overall lower mutation burden and less genomic instability in AYA patients compared to adult counterpart[40,41]. This may lead to lower immune recognition and hence reduced ICI efficacy. Other studies found similar mutational burdens between AYA and adults[12,18]. We observed a higher frequency of somatic mutations in *BRAF*, but not *PTEN*, in AYA patients compared to adults, and the AYA mutational profiles were not distinctly associated with TME profiles or ICI-resistance. This suggests that extrinsic factors perhaps play a greater role in ICI resistance in AYA melanoma, which warrants further investigation in larger AYA cohorts and using a more comprehensive genomic sequencing panel[42]. We observed a high frequency of *TERT* promoter mutations in AYA melanomas that showed a trend of co-occurrence with *BRAF* mutations, similar to findings in older adult patients which potentially suggests an intrinsic pathway of immunosuppression[43].

As the first comprehensive immunogenomic study of AYA melanoma to date, we demonstrated the potential for generating treatment-personalization models with multi-omic studies through the YIM score model and drug prediction analysis. Although limited by the current sample size for statistical strengths, the YIM score gene set displayed an over-representation of biological processes such as cell differentiation and cellular developmental process, which indicates potential tumor evolution into immune evasive subtypes[44]. The YIM score gene set also demonstrated the downregulation of immune-related signaling pathways such as antigen processing and presentation, natural killer cell mediated cytotoxicity and B cell receptor signaling, which supports our immunosuppressive pathway findings in ICI-resistant subtypes of melanoma, and together suggests that an immunogenic state is required for T cell infiltration, but further immune and TME interactions (perhaps involving B cells, antigen-presenting cells, and other immune and stromal cell types)[45] are required to sustain T cell resilience[46] and ICI response. Given the limited number of AYA patients who achieved durable ICI response in our study cohort, the non-resistant comparison group data could be influenced by features of acquired resistance. This was indicated by the similar immunophenogram profiles and comparable levels of T cell infiltration in the TME and between the non-resistant and SIL^high patients. Further studies of well curated ICI responsive AYA patients are needed to tease out the nuance of resistant groups and mechanisms of ICI response.

Published gene signatures for ICI response prediction, including IPRES[47] and various immune signatures[48–51], were predominantly based on the expression profiles of adult melanoma patients. We show that AYA patients have distinct immunogenomic profiles, and the YIM score demonstrated superior predictive value in our AYA cohort compared to published signatures. We recognize that the predictive value of the YIM score requires validation in independent cohorts. In addition, our study provides a preliminary framework for designing a treatment switch plan for ICI resistant AYA patients, guided by personalized biomarkers. Further investigation into synergistic drug combinations and their active molecular context will allow for the identification of clinically relevant drugs and combinations for patient groups with unique immunogenomic profiles. Through this study, we hope to propose a starting framework that would enable the identification of AYA patients at high risk of ICI resistance, and encourage the generation of more AYA cancer data that will empower future research to allow biomarker-driven clinical guidance and molecularly-informed enrichment of personalized treatments.

## Methods

References for analysis algorithms used as described below are summarized in Source Data.

### Patient samples and treatment

The research protocol was approved by the institutional review boards and ethics committees (Sydney Local Health District Human Research Ethics Committee, Protocol No. X-15-0454 and HREC/11/RPAH/444).

The Melanoma Research Database of the Melanoma Institute Australia (MIA) was used to identify cohort 1 AYA patients ($n = 28$) with the following criteria: (i) patients must have had AJCC[52] stage III or IV melanoma, (ii) ≤30 years of age at primary diagnosis and (iii) received treatment for melanoma with any of the following ICI therapies, including anti-PD-1 (nivolumab or pembrolizumab) monotherapy, anti-CTLA-4 (ipilimumab) monotherapy, and combination of anti-PD-1 and anti-CTLA-4 antibodies (ipilimumab combined with nivolumab or pembrolizumab) as detailed in Fig. 1 and Supplementary Data 1. Patients who received the treatments as a part of a clinical trial are detailed in Supplementary Data 1, all patients have ceased trials except AYA21 who is currently enrolled in the trial for survival follow-up (treatment completed). Patients who received ICI treatments in both the adjuvant and advanced setting were included. For patients treated in the advanced setting, response was determined as per the RECIST 1.1 criteria[53] and classified as complete response (CR), partial response (PR), stable disease (SD) and progressive disease (PD). Seven patients (AYA 4, 5, 6, 22, 25, 26, 28) were treated with adjuvant ICI. All biospecimens were obtained from the MIA Biospecimen Bank, with all patients providing written informed consent. All samples were

pathologically assessed prior to inclusion into the study as previously described[32].

Cohort 2 and 3 were obtained from published datasets (Fig. 1a, Supplementary Data 1). Cohort 2 comprised of 71 adults (>30 years of age at diagnosis) with advanced melanoma from a previous study[15], their transcriptome and multiplex immunofluorescence data were analyzed in this study. Cohort 3 comprised of 19 AYA patients from The Cancer Genome Atlas-Skin Cutaneous Melanoma (TCGA-SKCM) project, these patients do not have accessible clinical treatment record, their transcriptome data were used for the baseline comparison of immune profiles (CIBERSORT deconvolution) between AYA (total $n = 47$) and adult patients.

Patients treated with ICI who achieved a best RECIST response of CR, PR, or SD lasting over 6 months prior to progression in the advanced setting or a recurrence-free survival of over 12 months (since treatment initiation) in the adjuvant setting were deemed "non-resistant" to ICI. Conversely, patients who did not meet these criteria were classified as "ICI-resistant".

### Nucleic acid extraction

Both DNA and RNA were extracted from 10 µm-thick FFPE tissue sections using the AllPrep DNA/RNA FFPE Kit (Qiagen) according to the manufacturer's instructions. After deparaffinization and tissue lysis, the lysate was centrifuged to produce RNA-containing supernatant and DNA-containing pellet, which subsequently underwent separate purification procedures. Total RNA and DNA were eluted in 14–30 µl of RNase-free water and 30–100 µl of ATE buffer, respectively. Samples were quantified using the Qubit Fluorometer (Thermo Fisher Scientific) and were further assessed for DNA quality using a quantitative PCR. The quantity of amplifiable DNA in samples are evaluated with the Archer DNA QC assay to determine the amount of DNA input for making sequencing libraries.

### Next-generation custom amplicon sequencing

All recurrently mutated coding and non-coding genes discovered in our previous whole genome sequencing of a large cohort of melanomas[18,32] were designed into a custom next-generation sequencing panel in the ArcherDX Assay Designer (Supplementary Data 10). Sequencing libraries were prepared and barcoded using the unique molecular identifier and index tagging following the VariantPlex Somatic Protocol (ArcherDx). Pool-library was loaded at 1.2 pM concentration with 20% PhiX and sequencing was performed using the NextSeq 500 Illumina sequencer using 300 cycle high output reagent kit.

### DNA sequence data processing and variant calling analysis

The Illumina Local Run Manager Analysis Service v3 was used to generate FASTQ files and processed using the VariantPlex Archer Analysis Pipeline v6.2.7 with default settings, except for down-sampling default option which was disabled (changed to 0) to ensure use of all read pairs. Sequences were aligned to the hg19 reference genome and somatic variants were called using Freebayes and Lofreq and duplicate calls merged in the Archer Analysis 6 software. The quality of the sequencing data was assessed as the number of unique start sites (from DNA and ambiguous reads) calculated per GSP2 (across the entire panel). Samples with a sequence or variant score of <50 for average unique DNA were excluded from further analysis (AYA 3, AYA6, AYA8, AYA18) as per the manufacture's guidelines. Subsequent variants were filtered using the sequencing metrics of the annotated variant call file as follows: (1) UDP > 10, UAO > 6 (except *TERT* promoter), DAO > 1 (except *TERT* promoter), (2) HomopolymerCount<6, (3) gnomAD AD < 0.01 and Global AF < 0.01, (4) No positive call for sequence direction bias or Sample strand bias, (5) Sequence Direction Strand Bias Probability > 0.1 (except *TERT* promoter), (6) If the call was made with FreeBayes, then Sample Strand Bias Probability > 0.05 or Sample

Strand Bias Ratio >0.4, as well as 95MDAF < 0.2, (7) If the call was made with LoFreq, AF > 0.1 and AF < = 0.2, and (8) For protein-coding mutations, only those with a High or Moderate impact based on Ensembl predicted consequences were kept. For non-coding mutations, only those with a sequence ontology term of 5_prime_UTR_variant or upstream_gene_variant consequence were kept. Remaining variants were annotated using the Cancer Genome Interpreter to predict the functional significance of the variants and produce the mutation annotated file (MAF). MAFs were analyzed and visualized with maftools v2.8.05.

### Whole-transcriptome sequencing

The mRNA samples were fragmented in preparation for cDNA synthesis and library construction using the TruSeq RNA Exome Library Prep Kit (Illumina) according to the manufacturer's protocol. Library quality was assessed on an Agilent 2100 Bioanalyzer using a High Sensitivity DNA chip prior to paired-end sequencing on the Illumina NovaSeq 6000 machine, and a median of 87 million reads per sample was generated.

### Whole-transcriptome data processing

The data processing pipeline for whole transcriptome sequencing was described previously[15]. Briefly, after removing the adapter sequence in Trimmomatic, the sequences were aligned to Ensembl GRCh38 *Homo sapiens* reference genome using TopHat v2.0.8. Sequence alignment and sorting were performed using Bowtie v2.1.0 and SAMtools v0.1.19. Alignment statistics of aligned reads were generated using FastQC and RNA-SeQC for quality control, and sequences with a score >Q35 (>99.9% base call accuracy) were selected for analysis.

### Differential gene expression analysis

Gene count and differential expression analysis was performed using HTSeq and DESeq2 v1.32 with default parameters. After filtering genes with low counts of ≤20 counts for ≤5 samples, the genes were subsequently normalized using the *DESeq* function. Differentially expressed genes (DEGs) are identified as those with an adjusted *P* value of <0.05 with Benjamini-Hochberg multiple testing correction.

### Calculating cell proportion and immunophenotypes based on gene expression profiles

Normalized gene counts were used to determine the relative proportions of immune cells and the types of major determinants relating to tumor immunological features using CIBERSORT and Immunophenogram[29] respectively. The CIBERSORT was performed using the LM22 gene signature, with the default parameters applied in relative mode – B-mode batch correction, and the number of permutations set to 100 generating Pearson correlation coefficients. The algorithm obtained from immunophenogram calculates mean expression z-scores for marker genes of immune cells and features (effector cells, suppressor cells, checkpoints, and major histocompatibility complex) across each sample[29]. The immunophenograms were used to stratify AYA patients for downstream analyses.

### Gene set enrichment and pathway analysis

The curated gene sets including Kyoto Encyclopedia of Genes and Genomes (KEGG) and Gene Ontology (GO; including biological pathways [BP], cellular compartments [CC], and molecular functions [MF]) were downloaded from the Molecular Signature Database (MSigDB). Gene set enrichment analysis was performed using the GSEA() function in clusterProfiler v4.0.5 based on the log2 fold change of DEGs, and enriched signaling pathways were identified with an adjusted *P* value < 0.05. The biological features (including biological processes, molecular functions and pathways) and their corresponding genes were visualized with clusterProfiler v4.0.5.

## Gene signature scores

The gene signature enrichment scores for predicting poor immunogenicity in AYA melanomas were calculated using non-parametric, rank-based scoring method implemented in singscore:

$$S_{dir,i} = \left( \frac{\sum_g R^g_{dir,i}}{N_{dir,i}} \right)$$

Where:

- $dir$ is the gene set direction (i.e., expected up- or down- regulated genes in AYA ICI-resistant melanoma);
- $S_{dir,i}$ is the score for sample i against the directed gene set in AYA patients;
- $R^g_{dir,i}$ is the rank of gene $g$ in the directed gene set (increasing transcript abundance for expected up-regulated genes and decreasing abundance for expected down-regulated genes in ICI-resistant AYA melanoma);
- $N_{dir,i}$ is the number of genes in the expected up- or down- regulated gene set that are observed within the data (i.e., YIM genes not present within the sample RNA data are excluded).

Differential gene expression analysis was performed between ICI-resistant AYA patients and non-resistant CR patients (no primary resistant and did not progress), PR patients were removed due to heterogenous biological features and ICI outcome (short PFS). For the singscore calculation, YIM genes upregulated in ICI-resistant patients were positively weighted (+1) and genes upregulated in CR patients were negatively weighted (−1); IPRES genes were weighted equally (+); and for the IMMU score, immune effector genes (chemokine, interferon-γ, T effector, and T cell inflamed signatures) were positively weighted and immunosuppressive genes (B-catenin, TGFβ and immunosuppression signatures) were negatively weighted. Tumor immune dysfunction and exclusion (TIDE) computational model was used for calculating T cell dysfunction scores and performing biomarker evaluation of the YIM signature[28].

## Identification of druggable target genes

To select the druggable genes, we obtained a panel of genes (Supplementary Data 12) based on the significantly enriched signaling pathways as described in the previous section, and crossmatched with the Drug-Gene Interaction Database (DGIdb), similar to the drug-target network analysis done in a previous study[40]. Actionable drug targets were identified by filtering for those with "approved", "antineoplastic" and "immunotherapy" drugs in the DGIdb. The potential target genes and treatment switch plan were illustrated using networkD3.

## Multiplex immunofluorescence staining

Whole slide mIF staining was performed on 4 µm FFPE tissue sections mounted on Superfrost Plus slides (Thermo-Scientific). Panels were optimized and stained as previously described[15,54]. Briefly, slides were first heated at 65 °C for 30 min, deparaffinized in xylene and rehydrated in decreasing concentrations of ethanol. Slides were stained in the intelliPATH FLX® Automated Slide Stainer (Biocare Medical) using the tyramide signal amplification (TSA) method. In this method, tissue sections underwent antigen retrieval at 110 °C for 10 min and blocking with hydrogen peroxide before incubation with primary antibodies (CD3 1:1500, CD8 1:1500, FOXP3 1:250, ICOS 1:2000, CD39 1:1000, Grazyme B 1:200, PD-1 1:1500, SOX10 1:200) followed by signal amplification and Opal visualization. Antigen retrieval was repeated after each staining round and after the final staining round slides were counterstained with spectral DAPI (Akoya Biosciences, 1:2000). Slides were coverslipped in Prolong Diamond Antifade Mountant (Thermo-Scientific). Using this method, all samples were stained and visualized for the regulatory and CD8 T cell characterization panels as detailed in Supplementary Data 13.

## Image acquisition

Stained sections were imaged using the Vectra 3.0.5 Automated Quantitative Pathology Imaging system (Akoya Biosciences) using the FITC, Cy3, Texas Red, Cy5 and DAPI filters. 20X high power field (HPF) images covering the entire tumor were acquired and spectrally unmixed using inForm v2.4.2 (Akoya Biosciences). HPFs were subsequently stitched together for further analysis using HALO v3.2 (Indica Labs).

## Image analysis

Whole tissue sections were included for quantification and downstream analysis, which included the tumor core and the immediate surrounding stromal (peritumor) region. The tumor-stroma boundary was manually annotated by XB. Areas of necrosis, tissue folds and staining artifacts were excluded from analysis. A machine learning (random forest) based classifier algorithm was trained on images from the study to classify tissue as either tumor or peritumor. Cells were recognized based on nuclear DAPI and/or SOX10 staining. Positivity thresholds were manually optimized for each marker based on mean cellular intensity (nuclear or cytoplasmic). Cell populations were phenotyped based on the expression of individual markers, e.g., T cell: CD3⁺SOX10⁻.

## Statistical analysis

Where indicated, data were analyzed for statistical significance and reported as $P$ values. Continuous data were analyzed by nonparametric Mann–Whitney U test when comparing two independent groups and two-way ANOVA when comparing more than two groups. Categorical data were analyzed by Fisher's test for contingency tables with two rows and two columns, and by chi-square ($\chi^2$) test for contingency tables with more than two rows or columns. $P < 0.05$ was considered statistically significant.

Kruskal-Wallis test, corrected with the Benjamini-Hochberg method, was used to compare AYA versus adult CIBERSORT immune cell proportions; and multiple Mann–Whitney test corrected with the two-stage step-up method of Benjamini-Krieger-Yekutieli was used to compare Group 1 vs Group 2 mIF cell densities, as this correction method is more compatible when comparisons are negatively correlated (e.g., if T$_{reg}$ is high, CD8 T cell is low).

Evaluation of survival patterns between different response groups of AYA patients was performed by the Kaplan–Meier method, and results were ranked according to the Mantel–Cox log-rank test. $P < 0.05$ was considered statistically significant.

To evaluate the performance of YIM score, areas under the curve (AUC) for receiver operating characteristic (ROC) curves were calculated and visualized using pROC v1.18.

## Reporting summary

Further information on research design is available in the Nature Portfolio Reporting Summary linked to this article.

## Data availability

The RNA sequencing data for cohort 1 MIA AYA patients (n = 28) is deposited in the European Nucleotide Archive (ENA) under accession number PRJEB52880. DNA sequences of cohort 1 MIA AYA patients (n = 28) are available at the European Genome-Phenome Archive under accession number EGAS50000000238. The GRCh38 *Homo sapiens* reference genome was downloaded from Ensembl database (https://www.ensembl.org/Homo_sapiens/Info/Index). External datasets analyzed include the sequencing data of AYA patients from TCGA-SKCM project (https://portal.gdc.cancer.gov/projects/TCGA-SKCM), and adult from ENA (PRJEB23709). Data for the drug prediction analysis were retrieved from the DrugBank database (v5.1.3) [https://go.drugbank.com/releases/5-1-3]. All other data that support the findings of this study are available upon reasonable request from

the corresponding authors. Melanoma Institute Australia will promptly review all data requests to ensure that intellectual property and confidentiality obligations are met. A Material Transfer Agreement will be used to transfer any and all data that can be shared. Source data are provided with this paper.

## Code availability

Codes for the bioinformatics analysis of key results in this paper are available at the AYA_Melanoma Github repository (https://github.com/cameliaquek/AYA_Melanoma) and on Zenodo[55].

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

## Acknowledgements

This work was supported by Melanoma Institute Australia, the University of Sydney Medical Foundation, Cancer Institute NSW and National Health and Medical Research Council of Australia. X.B., G.H.A. and K.J.N. are supported by the University of Sydney and Melanoma Institute Australia Scholarships. G.V.L., R.A.S and A.M.M. are supported by a individual NHMRC Investigator Grants. J.S.W. is supported by an NHMRC investigator fellowship (APP1174325). A.M.M. is also supported by Nicholas and Helen Moore and Melanoma Institute Australia. C.Q., T.N.G. and I.P.DaS. are supported by the Cancer Institute NSW (2020/ECF1153, 2020/ECF1244, and 2021/ECF1376). Support from The Cameron Family, The Ainsworth Foundation, Charles Perkins Centre Seed Fund, Tour De Cure, the CLEARBridge Foundation as well as from colleagues at MIA and Royal Prince Alfred Hospital is also gratefully acknowledged.

## Author contributions

G.V.L., R.A.S., J.S.W. and C.Q. conceived and designed the study. K.J.N., I.P.D.S., M.S.C. and A.M.M. collected the clinical data and samples. P.M.F. performed pathological examination of the tissue samples. X.B., G.H.A., T.N.G. and P.S. conducted the experiments. X.B., C.Q. and I.A.V. performed the data analysis. U.P. reviewed the immune cell phenotypes. X.B. drafted the manuscript. All authors contributed to the revision and editing of the manuscript.

## Competing interests

G.V.L. is consultant advisor for Agenus, Amgen, Array Biopharma, AstraZeneca, Bayer, BioNTech, Boehringer Ingelheim, Bristol Myers Squibb, Evaxion, Hexal AG (Sandoz Company), Highlight Therapeutics S.L., IO Biotech, Immunocore, Innovent Biologics USA, Merck Sharpe & Dohme, Novartis, PHMR Ltd, Pierre Fabre, Regeneron, SkylineDx B.V., and Scancell. R.A.S. has received fees for professional services from SkylineDx B.V., IO Biotech ApS, MetaOptima Technology Inc., F. Hoffmann-La Roche Ltd, Evaxion, Provectus Biopharmaceuticals Australia, QBiotics, Novartis, Merck Sharp & Dohme, NeraCare, Amgen, Bristol Myers Squibb, Myriad Genetics, and GlaxoSmithKline. M.S.C. is consultant advisor for Amgen, Bristol Myers Squibb, Eisai, Ideaya, Merck Sharp & Dohme, Nektar, Novartis, Oncosec, Pierre Fabre, QBiotics, Regeneron and Roche, and honoraria for Bristol Myers Squibb, Merck Sharp & Dohme, and Novartis. A.M.M. is on the advisory board for Bristol Myers Squibb, Merck Sharp & Dohme, Novartis, Roche, Pierre Fabre, and QBiotics. I.P.S. is on the advisory board for Merk Sharp & Dohme, and has received fees for professional services from Roche, Bristol Myers Squibb, and Merck Sharp & Dohme. The other authors declare no competing interests.
