## [Peer Review File · Nature Communications]

Stroma-infiltrating T cell spatiotypes define immunotherapy outcomes in adolescent and young adult patients with melanomaREVIEWER COMMENTS

Reviewer #1 (tumour heterogeneity and spatial organisation):

In this study, the authors have looked at the responses of adolescent and young adult (AYA) melanoma patients to immune checkpoint blockade, primarily through RNAseq analysis and spatially segmented immunofluorescent staining of a cohort of 28 AYA melanoma patients from MIA. The strengths of this study are the unique ICI-treated AYA melanoma cohort the authors have assembled, and the insights into the TME of AYA melanoma patients as distinct from adult melanoma patients. The authors highlight that previous studies of age in patients in melanoma have had a range of age cut-offs, but frequently stratify patients as younger/older than 50 or 65, whereas the cohort assembled here is truly young patients, ages 16-30. This study thus fills a meaningful gap in data available in the literature. The authors show that there is a dramatically increased infiltration of Tregs into the peritumoral space of AYA melanoma patients, which is absent in older patients, and as such AYA patients have a unique TME that likely affects their response to ICIs.

However, there are also several points in this study where the data either needs to be strengthened or the description of the data needs to be more nuanced, as the authors currently come across as overinterpreting their data in several places. In particular, the comparison of ICI-resistant and non-ICI-resistant AYA patients has some gaps, as does the interpretation of the significance of "SIL-high" and "SIL-low" spatiotypes. The major points of critique fall under this overall theme, and are as follows:

Major points:

1. The authors make the statement "ICI-resistant AYA melanomas harbored a higher density of Tregs compared to non-resistant and resistant adult melanomas" (lines 192-193) is misleading. Non-ICI-resistant AYA melanomas also had high densities of Tregs, and so high Treg density is not uniquely a feature of ICI-resistant AYA melanomas as this sentence seems to imply. The authors' data show that there are higher Tregs in all AYA melanomas compared to all adult melanomas, but there is no evidence that these are increased in resistant AYA melanomas or definitively related to resistance.
2. Related to this point, the text for Fig 3c states the density of Tregs in ICI-resistant AYA patients is higher than in non-ICI-resistant AYA patients (line 241), but this is not supported by the significance test ($p = 0.28$). Combined with comment #2 above, it comes across that the author's wish to claim that Treg density is associated with resistance to ICI in AYA patients, but it does not seem the data is sufficiently clear to prove this. The authors either need to provide stronger data to support this conclusion (e.g., a statistically significant difference in Tregs between resistant and non-resistant patients) or rephrase the text.
3. The authors classify ICI-resistant patients as "SIL-high" or "SIL-low". It would strengthen this classification scheme and its potential clinical utility quite a bit if the authors were to also assign non-ICI-resistant patients to the SIL-high and SIL-low classifications, and then directly ask if the SIL-high profile is associated with better response/less resistance than SIL-low. The authors show data that hints this might be the case (SIL-high patients have slightly better survival, and have GzmB+ CD8 T cells at similar levels to non-resistant patients), but they currently stop short of testing whether identifying patients as "SIL-high" pre-treatment (e.g., in resistant and non-resistant patients, since resistance would not yet be known) is associated with better outcomes for those patients. This analysis should be done, and would greatly strengthen the impact of the SIL-high vs SIL-low classification. Currently, the meaning/implication of the SIL-high versus SIL-low classification the authors have developed is unclear, as the conclusions about mechanisms of resistance in each group are speculative and not tested in this paper. Further, in the discussion section the authors raise the idea that SIL-low patients may not be good candidates for ICI (lines 415-422), however the case for these conclusions would be a lot stronger if the authors could show that SIL-low spatiotype predicts poor responses to ICI. If this is the case, presumably the SIL-low would be poorly represented in the "non-resistant" patients. It seems the authors already have the data that should enable this classification for non-resistant patients to be done.

4. In Fig. 3i-j – The authors show a Kaplan-Meier curve comparing OS and PFS for non-resistant, SIL-high, and SIL-low patients. Highly significant p-values are given, but it is not clear what is being compared in the significance test. Each of the 3 arms should be compared separately (non-resistant vs SIL-high, non-resistant vs SIL-low, and SIL-high vs SIL-low) and significance provided for each of these three comparisons. In particular, a statistic for the difference between the SIL-high and SIL-low groups should be included, since the point the text is making is that these two groups have meaningfully different outcomes. As presented, there is some lack of clarity as to whether the high significant p-values are being driven by the “non-resistant” survival rates, which are—not surprisingly—much better than those of both cohorts of “resistant” patients.

5. The authors’ motivation for developing YIM score for predicting resistance in AYA patients is clear and it is a worthwhile endeavor, however it seems this score has been both developed on and test on the same (small) cohort of 28 AYA patients. It is almost a given that a scoring algorithm will perform well on the dataset it was developed on/trained on, and so the success of this algorithm in predicting responses of their cohort is not all that meaningful. To establish utility of the YIM score, the authors need to test its performance on another independent dataset. Recognizing that there are not a lot of datasets available for ICI-treated AYA melanoma patients, it is still problematic to claim the YIM score has any value without additional external validation. Potentially the authors could divide their own dataset into training and testing cohorts, but the number of patients they have may be too small for this to be feasible, and this would also involve altering the current score algorithm.

6. The authors perform a drug gene interaction prediction, in which they identify “druggable targets” in AYA patients and subsequently propose a “treatment switch” plan for patients. This analysis is problematic, and the conclusions the authors draw from it are extremely underpowered and unsubstantiated. The upregulation of a pathway does not necessarily mean that its inhibition is going to lead to a meaningful impact on disease. The rest of this paper has gone to considerable lengths looking at how one particularly drug class/combination (ICI) has different responses in AYA versus adult patients, and it is inconsistent with this entire premise to propose that KEGG pathway analysis is a good basis for developing personalized treatment plans. For Fig. 4d, the analysis performed should be clearer (What are the genes listed in the middle? Do these represent upregulated genes or upregulated pathways or mutant genes? What does the thickness of each colored line mean?). For Extended Fig. 5f and the text that relates to it, this diagram and its entire premise seems inappropriate as its conclusions (e.g., that Group 1/SIL-high patients should be put on an alternative ICI) are speculative and as of yet completely unsubstantiated by data. The text should reflect the speculative nature of these ideas (e.g., that SIL-high versus SIL-low status might indicate what type of follow-up treatment would be most successful after a patient fails ICI), rather than presenting them as a conclusion of the data in this paper, and this figure in its current form does not seem appropriate to include.

Additionally, a few minor points the authors should also address:

1. The description of the two AYA cohorts is confusing. The authors introduce their AYA cohort as a cohort of 47 patients, however from the flow chart, this is really 2 cohorts. The first is a cohort of 28 patients from MIA who were treated with ICI, whom the majority of the data in this paper is based on. Additionally, there is a cohort 19 untreated patients whose RNAseq data was pulled from TCGA—it is actually not clear where in the paper this data is used, if at all. The authors should introduce these two AYA cohorts more clearly and where data from each is used. (Or alternatively, just introduce AYA cohort 1 and the adult cohort 2 upfront, and mention the TCGA patients later in the paper at the point their data is used.)

2. On line 144, it says “Of the X AYA patients...” – what is X?

3. The authors make the statement that ICI response rates were lower in AYA patients than adults. This is true when aggregating combination and monotherapy treated patients (responses are 38% in AYA and 63% in adults), but it seems like the data is actually much more nuanced. From my understanding of the breakdown of responses later in the paragraph discussing these results, it looks like 78% of AYA patients respond to combination treatment while 68% of adults do. I don’t

know if this ends up being a statistically significant difference that AYA patients respond better to combo ICI, but it is not at all consistent with the idea that ICI generally works worse for AYA patients. It seems like the conclusion that ICI responses are worse in AYA patients is driven only by responses monotherapy (not clear if it is true for both anti-CTLA-4 and anti-PD-1, or just one of those). This should be clarified in the text—the monotherapy versus combo therapy dichotomy is a very interesting finding itself, but it does not seem appropriate to generalize to say all ICI is less effective in AYA patients, unless there is additional data to support his claim.

4. The images in Fig. 2 are very poor quality, and make it difficult to evaluate the staining. In addition to the panels shown, it would also be useful to see representative images for the Foxp3 channel for AYA and adult patients, illustrating the point that AYA melanomas typically had significantly more Tregs and supporting the quantifications shown in Fig. 2f. The Foxp3 insets shown make it look like the adult patient actually has more Tregs, although this is quite difficult to see with the current quality of the images.

5. In Fig. 3c, the figure says “peritumor Tregs” while the text just says “Tregs” – this should be clarified which is being shown in Fig. 3c. If Fig. 3c is showing just peritumor Tregs, did the authors look at intratumoral Tregs as well?

6. In Extended Fig. 4c-d, there are p-values listed to the right of Fig. 4d. Are these p-values for the peritumoral cell densities (Fig. 4d) or also for the intratumoral cell densities (Fig. 4c)? A p-value for intratumoral densities should be included separately for Fig. 4c.

7. It would be nice to cell quantifications for non-resistant tumors included in the bar graphs in Extended Fig. 4c-d. Since this data is included in Fig. 3h for Gzmb+ CD8s, it seems the authors have collected this data, and it would be useful to see it included here and to be able to compare T cell infiltrates in non-resistant tumors to each of the two resistant subtypes. It would be really interesting to see whether “SIL-high” tumors (which have high Gzmb+ CD8s) generally have a T cell infiltrate similar to non-resistant tumors—this would be impactful, particularly if it can be coupled with data showing what fraction of non-resistant patients have tumors that look like SIL-high versus SIL-low resistant patients.

8. Text discussion of figures is not sequential, which is confusing. The text description of Extended Fig. 2e-f comes before Fig. 2c-d; and Fig. 3g comes before Fig. 3e-f

Reviewer #2 (melanoma in adolescents and young adults):

This is an important study which demonstrated a peculiar immunogenomic profile in adolescent and young adult melanoma patients. Moreover the study individualized specific tumor microenvironment features in immune checkpoint inhibitors-resistant adolescent and young adult melanoma.

The work is original and the conclusions are consistent with the experimental data. The methodology is appropriate and the study is very well structured.

I would suggest to include treatment flow chart based on the transcriptomic, genomic, and spatial cellular profiles of the tumor microenvironment in adolescent and young adult melanoma.

Reviewer #3 (skin cancer and physiology, single-cell transcriptomics):

In this manuscript, Bai et al. compared the transcriptomic, genomic, and intratumoral immune landscapes of the tumor microenvironment (TME) of melanomas from adolescent and young adult (AYA) and adult patients to assess the pathways associated with immune checkpoint inhibitor (ICI) resistance in AYA patients, and to define biomarkers that allow treatment response prediction. Since the incidence of melanoma in young patients is significantly higher compared to older patients, and contradictory data about frequency of resistance to ICI treatment with age as a

determining factor were reported, this manuscript addresses a timely question.

The authors revealed that in general, response to ICI treatment and survival were poor among AYA patients treated with ICI. Their results suggest that AYA melanoma patients benefit from combination ICI treatment as far as response and survival are concerned. Furthermore, they identified 2 ICI-resistant subgroups of AYA melanoma which display differences in stroma infiltrating lymphocytes (SILs) compared to adult TME; SIL^{high} are enriched for Tregs (peritumorally) and present with alternative checkpoint molecule expression, while the SIL^{low} tumors showed a lack of immune induction. They also established an immunosuppressive score for AYA melanoma patients (YIM score) which predicted ICI response for the analyzed AYA cohort. They also proposed potential immune and non-immune based therapeutic targets for personalized therapies of ICI-resistant AYA patients.

This is a descriptive study with most of the data generated through computational analysis of sequencing data. In general, the analysis pipeline of the datasets follows previously published computational approaches and seems to be well done. The authors performed CIBERSORT immune cell deconvolution on pre-treatment melanoma biopsy bulk-seq data to assess the tumor immune microenvironment, and confirmed some of their findings (increased Treg numbers in AYA patients, especially in peritumoral regions) by immunostainings.

They also established a gene-expression-based immunosuppressive score for AYA melanoma patients (YIM score) which predicted ICI response for the analyzed AYA cohort. Furthermore they performed a drug gene interaction prediction analysis and suggested potential immune and non-immune based therapeutic targets for personalized therapies of ICI-resistant AYA patients.

Major comments:

Fig. 2d-e: "we performed immunofluorescence (mIF) staining (Fig. 2d-e; Extended Data Fig. 2b) for Treg populations in the matched tumor". What exactly is the matched tumor? I assume it is the pre-treatment biopsy but it is not clearly stated if it is before or after treatment? Furthermore, cellular composition varies in different regions of a tumor. Was the whole tumor tissue used for quantification? If not, what were the selection criteria for the analyzed regions of interest?

Fig. 2/ Extended Figure 2: Are all AYA patients with high Foxp3 cell numbers resistant to ICI therapy?

Is there a correlation between BRAF mutations and ICI resistance in AYA patients, or more specifically within SIL^{high} and SIL^{low} groups? According to Extended Fig. 7 most SIL^{low} patients have BRAF mutations while SIL^{high} have not.

Do BRAF mutations correlate with Treg increases?

Fig. 3c/ line 240-242: "The density of Treg cells were higher in the AYA ICI-resistant patients compared to the AYA non-resistant patients". There is a trend but no significant difference. Please rephrase!

Fig. 3: Is there a SIL^{low} group with a general lack of immune induction in the TME of adult resistant melanomas?

Line 406/407: "Our study showed that combination ICI therapy had better progression-free survival than monotherapy in AYA melanoma patients, particularly those with Treg enrichment". Where is the last part shown?

Apart from higher proportions of Tregs in AYA melanomas, and higher numbers of naïve B cells and lower numbers of plasma cells and M2 macrophages were reported in AYA melanomas. Is there a correlation between these cell types with ICI resistance or the SIL^{high} and SIL^{low} group?

Fig. 4c: no one can read the genes listed in the heatmap. It would make sense to only show the top 20 up- or downregulated genes or to zoom into some regions with genes of interest

Fig. 4: Was the YIM score calculated from AYA patients from cohort 1 and 3? If the score was calculated only with cohort 1, would the prediction fit to cohort 3 as a validation?

Taking into account that you find SIL^{high} and SIL^{low} groups within the AYA patients, would it make sense to calculate 2 separate YIM scores for each group? Furthermore, could the SIL-low score also predict ICI resistance in adult melanoma patients?

Minor comments:

The labels of some figures are very small and difficult to read, e.g. Figure 3e,f, Figure 4c and similar plots in the extended data.

Fig. 4: legends for b and c have been swapped;

RESPONSE TO REVIEWERS' COMMENTS

We would like to thank the reviewers for their thorough review and informed suggestions. We believe that the changes made in response to the reviewer comments have strengthened the work. Please find the point-by-point responses to the reviewers' comments below. Manuscript changes in response to the reviewer's comments are outlined below, and highlighted in the revised manuscript.

REVIEWER 1 (expert in tumour heterogeneity and spatial organisation of tumour-specific immune responses):

In this study, the authors have looked at the responses of adolescent and young adult (AYA) melanoma patients to immune checkpoint blockade, primarily through RNAseq analysis and spatially segmented immunofluorescent staining of a cohort of 28 AYA melanoma patients from MIA. The strengths of this study are the unique ICI-treated AYA melanoma cohort the authors have assembled, and the insights into the TME of AYA melanoma patients as distinct from adult melanoma patients. The authors highlight that previous studies of age in patients in melanoma have had a range of age cut-offs, but frequently stratify patients as younger/older than 50 or 65, whereas the cohort assembled here is truly young patients, ages 16-30. This study thus fills a meaningful gap in data available in the literature. The authors show that there is a dramatically increased infiltration of Tregs into the peritumoral space of AYA melanoma patients, which is absent in older patients, and as such AYA patients have a unique TME that likely affects their response to ICIs.

However, there are also several points in this study where the data either needs to be strengthened or the description of the data needs to be more nuanced, as the authors currently come across as overinterpreting their data in several places. In particular, the comparison of ICI-resistant and non-ICI-resistant AYA patients has some gaps, as does the interpretation of the significance of "SIL-high" and "SIL-low" spatiotypes. The major points of critique fall under this overall theme, and are as follows:

Response: We thank the reviewer for their supportive comments recognising that the current study fulfils a meaningful gap in the existing literature for AYA melanoma patients undergoing immunotherapy treatment. Based on the reviewer's constructive feedback, we have undertaken new analyses of the existing data, resulting in the addition of figures, and substantially reworded the description and interpretation of the results in the manuscript to reflect nuance in the data, as detailed in the responses below.

Major points:

1. The authors make the statement "ICI-resistant AYA melanomas harbored a higher density of Tregs compared to non-resistant and resistant adult melanomas" (lines 192-193) is misleading. Non-ICI-

resistant AYA melanomas also had high densities of Tregs, and so high Treg density is not uniquely a feature of ICI-resistant AYA melanomas as this sentence seems to imply.

Response: We noted that a high T_{reg} density is not a unique feature of ICI-resistant AYA melanomas. However, a majority of our ICI resistant (88%; 14 out of 16) and non-resistant (91%; 10 out of 11) AYA patients have higher T_{reg} densities compared to the median T_{reg} density in adult melanomas (12.99 cells/mm²). To provide clarity, we have rephrased the sentence to clarify that high T_{reg} is not a feature exclusive to resistant AYA in the revised manuscript:

- **RESULTS, page 6:** Comparison between the response groups showed that both ICI-resistant and non-resistant AYA melanomas harboured higher densities of T_{regs} compared to resistant (median = 72.8 and 122 vs 9.38 cells/mm², $P = 0.0006$ and 0.001) and non-resistant (median = 15.1 cells/mm², $P = 0.0048$ and 0.0069) adult melanomas (Extended Data Fig. 2c).

This result precedes our later characterisation of distinct SIL groups in ICI resistant and non-resistant AYA melanoma, which showed different T cell densities in the tumour and surrounding stroma. The SIL^{high} ICI resistant tumours were immunogenic, but enriched for peritumour T_{regs} , and we showed that the pronounced enrichment of T_{regs} may dampen the inflammatory response and contribute to poor ICI outcome. The indirect inhibition of Tregs with anti-CTLA-4 combination immunotherapy may improve the clinical outcome for some patients, this is in line with retrospective data showing that young melanoma patients ≤ 30 years have better or trends towards better outcomes with combination ICI vs anti-PD-1 monotherapy (objective response rates of 57% vs 44%, PFS 13.8 vs 4.0 months, OS 82.1 vs 24.1 months) (Wong et al., 2023).

Reference:

Wong, S. K., Blum, S. M., Sun, X., Da Silva, I. P., Zubiri, L., Ye, F., Bai, K., Zhang, K., Ugurel, S., Zimmer, L., Livingstone, E., Schadendorf, D., Serra-Bellver, P., Muñoz-Couselo, E., Ortiz, C., Lostes, J., Huertas, R. M., Arance, A., Pickering, L., Long, G. V., ... Johnson, D. B. (2023). Efficacy and safety of immune checkpoint inhibitors in young adults with metastatic melanoma. *European journal of cancer (Oxford, England: 1990)*, 181, 188–197. <https://doi.org/10.1016/j.ejca.2022.12.013>

2. Related to this point, the text for Fig 3c states the density of Tregs in ICI-resistant AYA patients is higher than in non-ICI-resistant AYA patients (line 241), but this is not supported by the significance test ($p = 0.28$). Combined with comment #2 above, it comes across that the author's wish to claim that Treg density is associated with resistance to ICI in AYA patients, but it does not seem the data is sufficiently clear to prove this. The authors either need to provide stronger data to support this conclusion (e.g., a statistically significant difference in Tregs between resistant and non-resistant patients) or rephrase the text.

Response: The high P value is potentially due to small sample size. We have rephrased the sentence in the revised manuscript to reflect the findings more accurately.

- **RESULTS, page 9:** There was a non-significant, but numerically higher **peritumoural** T_{reg} density in the AYA ICI-resistant patients compared to the AYA non-resistant patients (Fig. 3c; median = 68 vs 102 cells/mm², P = 0.37).

3. The authors classify ICI-resistant patients as “SIL-high” or “SIL-low”. It would strengthen this classification scheme and its potential clinical utility quite a bit if the authors were to also assign non-ICI-resistant patients to the SIL-high and SIL-low classifications, and then directly ask if the SIL-high profile is associated with better response/less resistance than SIL-low. The authors show data that hints this might be the case (SIL-high patients have slightly better survival, and have GzmB+ CD8 T cells at similar levels to non-resistant patients), but they currently stop short of testing whether identifying patients as “SIL-high” pre-treatment (e.g., in resistant and non-resistant patients, since resistance would not yet be known) is associated with better outcomes for those patients. This analysis should be done, and would greatly strengthen the impact of the SIL-high vs SIL-low classification. Currently, the meaning/implication of the SIL-high versus SIL-low classification the authors have developed is unclear, as the conclusions about mechanisms of resistance in each group are speculative and not tested in this paper. Further, in the discussion section the authors raise the idea that SIL-low patients may not be good candidates for ICI (lines 415-422), however the case for these conclusions would be a lot stronger if the authors could show that SIL-low spatioype predicts poor responses to ICI. If this is the case, presumably the SIL-low would be poorly represented in the “non-resistant” patients. It seems the authors already have the data that should enable this classification for non-resistant patients to be done.

Response: We thank the reviewer for this constructive feedback and have incorporated the suggested analysis into the manuscript. We have added the AYA non-resistant patients into immunophenogram (Extended Data Fig. 4) and cell profile (Extended Data Fig. 5) comparisons (figures shown below), finding overall similar immune profiles between SIL^{high} and non-resistant tumours at baseline – indeed, SIL^{low} is poorly represented in the ICI non-resistant group. Applying the same stratification criteria (SIL^{low} patients transcriptionally downregulate immune cell signatures and checkpoints, particularly the downregulation of *PD1* and at least one other checkpoint marker), we identified 7 SIL^{high} and 4 SIL^{low} patients among the ICI non-resistant group. We have also included the survival analysis for all AYA patients, where patients are stratified by ICI response status as well as SIL category (Extended Data Fig. 5 d-e), and just SIL category regardless of ICI response (Extended Data Fig. 5 f-g).

- **RESULTS, page 12:** We also stratified the ICI non-resistant patients by SIL category, given the small cohort, SIL spatioype did not differentiate survival outcome in ICI non-resistant patients. However, the survival analysis showed significantly longer OS and PFS in the non-resistant SIL^{high} group compared to the resistant groups, as well as a longer PFS in the non-resistant SIL^{low} group compared to the resistant SIL^{low} group (Extended Data Fig. 5d-e).

- RESULTS, page 12:** When AYA patients were stratified by the SIL spatiootype regardless of ICI outcome, the SIL^{high} group showed a trend of longer OS and a near-significant longer PFS (P = 0.0504) compared to SIL^{low} group (Extended Data Fig. 5f-g). The predictive and prognostic powers of SIL spatiootype warrant further investigation in bigger cohorts.

Extended Data Fig. 4

a Non-resistant

b SIL^{high}

c SIL^{low}

Extended Data Fig. 4 Immunophenograms of ICI non-resistant and resistant subtypes of AYA melanoma
(a) Immunophenograms of ICI non-resistant patients. **(b)** Immunophenograms of Group 1 (SIL^{high}) ICI-resistant patients. **(c)** Immunophenograms of Group 2 (SIL^{low}) ICI-resistant patients. Genes, category acronyms, and colour legends are shown in the example Immunophenogram at the bottom right. The +/- sign after the gene name represents the weighting when calculating the immunophenoscore. Abbreviations: MHC – Major histocompatibility complex (antigen processing); CP – Checkpoints (immunomodulators); EC – Effector cells; SC: Suppressor cells.

Extended Data Fig. 5

Extended Data Fig. 5 Distinct cellular profiles and survival outcomes in ICI non-resistant, SIL^{high} and SIL^{low} subtypes of AYA melanoma. (a) Intratumoural T cell density comparisons between ICI non-resistant and resistant Group 1 (SIL^{high}) and Group 2 (SIL^{low}) AYA patients; bar and lines represent mean and standard deviation; P values of Mann-Whitney tests are shown; NS, not significant ($P > 0.05$). (b) Peritumoural T cell density comparisons between ICI non-resistant and resistant Group 1 (SIL^{high}) and Group 2 (SIL^{low}) AYA patients; bar and lines represent mean and standard deviation; P values of Mann-Whitney tests are shown; NS, not significant ($P > 0.05$). (c) Comparisons of CIBERSORT immune cell proportions between ICI non-resistant and resistant Group 1 and Group 2 AYA patients; bar and lines represent mean and standard deviation; P values of Mann-Whitney tests are shown; NS, not significant ($P > 0.05$). (d-e) Kaplan-Meier curves comparing the overall and progression-free survival between SIL^{high} and SIL^{low} subgroups of ICI resistant and non-resistant AYA patients. (f-g) Kaplan-Meier curves comparing the overall and progression-free survival between SIL^{high} and SIL^{low} groups of all AYA patients with recorded outcome.

4. In Fig. 3i-j – The authors show a Kaplan-Meier curve comparing OS and PFS for non-resistant, SIL^{high}, and SIL^{low} patients. Highly significant p-values are given, but it is not clear what is being compared in the significance test. Each of the 3 arms should be compared separately (non-resistant vs SIL^{high}, non-resistant vs SIL^{low}, and SIL^{high} vs SIL^{low}) and significance provided for each of these three comparisons. In particular, a statistic for the difference between the SIL^{high} and SIL^{low} groups should be included, since the point the text is making is that these two groups have meaningfully

different outcomes. As presented, there is some lack of clarity as to whether the high significant p-values are being driven by the “non-resistant” survival rates, which are—not surprisingly—much better than those of both cohorts of “resistant” patients.

Response: We have performed the statistic by comparing the survival curves between different groups using log-rank test. We have revised the updated manuscript and now added the pairwise comparison P values to Fig. 3i-j (as shown below). The result output showed that SIL^{high} patients had a significantly longer PFS compared to SIL^{low} patients (P = 0.0113) (**RESULTS, page 12**). Overall, the results demonstrate that the SIL^{high} and SIL^{low} ICI resistant patients have poorer overall survival compared to ICI non-resistant patients, but their SIL, or immune status in the TME seem to affect the time to disease progression on ICI treatment.

Fig. 3: Resistant hotspot of peritumoral Tregs and immunosuppressed spatioypes of AYA melanoma at pre-treatment baseline. (i) and progression-free (j) survival between AYA ICI non-resistant patients (n = 11), and Group 1 (n = 10) and Group 2 (n = 6) resistant patients.

5. The authors’ motivation for developing YIM score for predicting resistance in AYA patients is clear and it is a worthwhile endeavour, however it seems this score has been both developed on and test on the same (small) cohort of 28 AYA patients. It is almost a given that a scoring algorithm will perform well on the dataset it was developed on/trained on, and so the success of this algorithm in predicting responses of their cohort is not all that meaningful. To establish utility of the YIM score, the authors need to test its performance on another independent dataset. Potentially the authors could divide their

own dataset into training and testing cohorts, but the number of patients they have may be too small for this to be feasible, and this would also involve altering the current score algorithm.

Response: We acknowledge that proper validation of the YIM score requires a larger and independent dataset. However, it is very difficult to source well-curated immunotherapy treated AYA melanoma samples/datasets, and the current cohort is too small to be divided into different cohorts (i.e. discovery, training and testing). We thank the reviewer for recognising this limitation in data availability. Given the small sample size, we have approached the YIM score using a rank-based scoring algorithm that is independent of the sample size, and this scoring algorithm has been validated by our group (Mao et al., 2023) and others (Bhuva et al., 2019). We have now revised the Introduction, Results and Discussion sections to provide clarity, and to highlight the novel and preliminary nature of the current YIM score (as shown below).

- **INTRODUCTION, page 4:** We toned down the predictive value of the YIM score – “We identified the genomic, transcriptomic, and spatial cellular features of two major subtypes of ICI-resistant AYA tumors (stroma infiltrating lymphocyte (SIL) high and low tumor subtypes) and established an **immunosuppressive score for AYA melanoma patients which differentiated the ICI response in this cohort.**”
- **RESULTS, page 16:** We have now added, “Given the limited cohort, we used a rank-based gene scoring approach that is independent of sample size”.
- **DISCUSSION, page 19:** We have rephased sentences to reflect the premature nature of the YIM score and discuss its limitations: “Given the limited number of AYA patients who achieved durable ICI response in our study cohort, the non-resistant comparison group data could be influenced by features of acquired resistance. This was indicated by the similar immunophenogram profiles and comparable levels of T cell infiltration in the TME and between the non-resistant and SIL^{high} patients. Further studies of well curated ICI responsive AYA patients are needed to tease out the nuance of resistant groups and mechanisms of ICI response...We recognize that the predictive value of the YIM score requires validation in independent cohorts.”

References:

- Mao, Y., Gide, T. N., Adegoke, N. A., Quek, C., Maher, N., Potter, A., Patrick, E., Saw, R. P. M., Thompson, J. F., Spillane, A. J., Shannon, K. F., Carlino, M. S., Lo, S. N., Menzies, A. M., da Silva, I. P., Long, G. V., Scolyer, R. A., & Wilmott, J. S. (2023). Cross-platform comparison of immune signatures in immunotherapy-treated patients with advanced melanoma using a rank-based scoring approach. *Journal of translational medicine*, 21(1), 257. <https://doi.org/10.1186/s12967-023-04092-9>

- Bhuva, D. D., Foroutan, M., Xie, Y., Lyu, R., Cursons, J., & Davis, M. J. (2019). Using singscore to predict mutation status in acute myeloid leukemia from transcriptomic signatures. *F1000Research*, 8, 776. <https://doi.org/10.12688/f1000research.19236.3>

Recognizing that there are not a lot of datasets available for ICI-treated AYA melanoma patients, it is still problematic to claim the YIM score has any value without additional external validation.

We calculated the YIM score for cohort 3 TCGA AYA patients, and stratified them into YIM high and low groups based on median cut-off (Supplementary Table 8). Although this cohort has no accessible clinical treatment data (i.e. RECIST) to validate the prediction of immunotherapy outcome, we found a trend of better disease specific survival for patients with lower YIM scores (less immunosuppression). The below figure is added into the revised manuscript as Extended Data Fig. 6f:

- **RESULTS, page 16:** Lower YIM score further showed a trend of longer disease specific survival in TCGA AYA (cohort 3) patients (Extended Data Fig. 6f).

Extended Data Fig. 6 Differentially expressed genes in AYA patients can help to predict response and inform treatment strategies in ICI-resistant patients. (f) Kaplan-Meier curves comparing the disease-free survival of YIM score high vs low (using group median score as cut-off) TCGA AYA patients (cohort 3).

We have also revised the Discussion section in the updated manuscript (as shown below).

- **DISCUSSION, page 19:** “Published gene signatures for ICI response prediction, including IPRES⁴⁷ and various immune signatures^{48, 49, 50, 51}, were predominately based on the expression profiles of adult melanoma patients. We show that AYA patients have distinct immunogenomic

profiles, and the YIM score demonstrated superior predictive value in our AYA cohort compared to published signatures. **We recognize that the predictive value of the YIM score requires validation in large independent cohorts.** In addition, our study provides a preliminary framework for designing a treatment switch plan for ICI resistant AYA patients, guided by personalized biomarkers. Further investigation into synergistic drug combinations and their active molecular context will allow for the identification of clinically relevant drugs and combinations for patient groups with unique immunogenomic profiles.”

6. The authors perform a drug gene interaction prediction, in which they identify “druggable targets” in AYA patients and subsequently propose a “treatment switch” plan for patients. This analysis is problematic, and the conclusions the authors draw from it are extremely underpowered and unsubstantiated. The upregulation of a pathway does not necessarily mean that its inhibition is going to lead to a meaningful impact on disease. The rest of this paper has gone to considerable lengths looking at how one particularly drug class/combo (ICI) has different responses in AYA versus adult patients, and it is inconsistent with this entire premise to propose that KEGG pathway analysis is a good basis for developing personalized treatment plans.

Response: The drug target interaction analysis was performed using an analytical approach that was conceptually similar to the published work (Akhavanfard et al., 2020), as we attempted to address the clinical unmet question of “what can we do when AYA patients fail to respond to the standard of care ICI?”. It is not our intention to propose that KEGG pathway analysis is a good basis for developing personalized treatment plans. Using our multi-omics data, we present a framework for answering this question that could be adapted in future studies and translated into the clinic. We agree with the reviewer’s comment and have clarified what was performed in the figure legend, and toned down the language in the Result section of the revised manuscript as detailed in the responses below.

For Fig. 4d, the analysis performed should be clearer (What are the genes listed in the middle? Do these represent upregulated genes or upregulated pathways or mutant genes? What does the thickness of each colored line mean?).

We have revised the figure legend in the updated manuscript to describe the components of Fig. 4d:

- **Legend of Fig. 4, page 14:** (d) Sankey diagram highlighting the **alternative drug-gene interactions in AYA ICI resistant patients (Supplementary Table 12)**; left represents the **total number of druggable gene and mutational targets in SIL^{high} and SIL^{low} groups**, middle shows **targetable expressed or mutated markers**, right shows the **drug categories**; **line thickness represents the proportion within each group**, for example, line will have a thickness of 1 if one patient expressed the drug target; bi-specific T cell engager, BiTE.

For Extended Fig. 5f and the text that relates to it, this diagram and its entire premise seems inappropriate as its conclusions (e.g., that Group 1/SIL-high patients should be put on an alternative

ICI) are speculative and as of yet completely unsubstantiated by data. The text should reflect the speculative nature of these ideas (e.g., that SIL-high versus SIL-low status might indicate what type of follow-up treatment would be most successful after a patient fails ICI), rather than presenting them as a conclusion of the data in this paper, and this figure in its current form does not seem appropriate to include.

We have revised the writing in the updated manuscript to reflect the speculative nature of the model treatment switch plan (previous Extended Data Fig. 5f, now Extended Data Fig. 6g):

- **RESULTS, pages 16-17:** To assess the expression of potential therapeutic targets for ICI-resistant AYA patients, we performed drug gene interaction prediction using the transcriptomic and mutational profiles, which identified 27 actionable targets with currently approved oncolytic drugs (Fig. 4d, Supplementary Table 12). Given the lack of therapeutic progress for AYA cancer patients in the past decades, our data here shows the potential room for improvement. **All ICI-resistant AYA patients expressed at least one currently druggable target at baseline, and their SIL^{high} versus SIL^{low} status might indicate what type of follow-up treatment would be most successful after the patient fails ICI.** In line with their state of increased immune activation, a proportion of SIL^{high} patients expressed alternative ICI targets (including *LAG3*, *IDO*, *TIGIT*), and have the potential to benefit from bi-specific T-cell engagers (BiTEs); SIL^{low} tumours with overall dampened immunogenicity showed potential drug gene interactions for immune augmentation targets (*IL2RG*, *TNFRSF9*) and mutational targets (*BRAF*, *PTEN*, *NRAS*, *TP53*). More multi-omic profiling for AYA patients will facilitate the development of novel drugs and personalised treatment plans, **as a general example, we summarised our TME profiling and drug interaction findings for ICI-resistant AYA melanoma patients into a treatment switch plan to exemplify future guides for translational research and clinical trial designs** (Extended Data Fig. 6g)

Reference:

Akhavanfard, S., Padmanabhan, R., Yehia, L., Cheng, F., & Eng, C. (2020). Comprehensive germline genomic profiles of children, adolescents and young adults with solid tumors. *Nature communications*, 11(1), 2206. <https://doi.org/10.1038/s41467-020-16067-1>

Additionally, a few minor points the authors should also address:

7. The description of the two AYA cohorts is confusing. The authors introduce their AYA cohort as a cohort of 47 patients, however from the flow chart, this is really 2 cohorts. The first is a cohort of 28 patients from MIA who were treated with ICI, whom the majority of the data in this paper is based on. Additionally, there is a cohort 19 untreated patients whose RNAseq data was pulled from TCGA—it is actually not clear where in the paper this data is used, if at all. The authors should introduce these two AYA cohorts more clearly and where data from each is used. (Or alternatively, just introduce AYA cohort

1 and the adult cohort 2 upfront, and mention the TCGA patients later in the paper at the point their data is used.)

Response: We now clarified in the Results and Methods sections that the TCGA cohort was used in the CIBERSORT analysis to gain a more comprehensive understanding of the AYA TME at baseline, while the rest of the study focused on the immunotherapy-treated MIA AYA cohort (n=28):

- **RESULTS, page 4:** This study analyzed 47 AYA and 71 older adult patients with metastatic melanoma. **The AYA group includes 28 patients who received ICI therapy in either the adjuvant or advanced setting (cohort 1), and 19 patients sourced from TCGA database with no accessible clinical record of immunotherapy treatment (cohort 3).**
- **RESULTS, page 6:** To assess the immune landscape of the AYA and adult melanomas at baseline, **we included transcriptomic data from a further cohort of AYA patients (cohort 3, n = 19) sourced from the TCGA database.**
- **METHODS, page 20:** Cohort 2 and 3 were obtained from published datasets (Fig. 1a, Supplementary Table 1). Cohort 2 comprised of 71 adults (>30 years of age at diagnosis) with advanced melanoma from a previous study¹⁵, their transcriptome and multiplex immunofluorescence data were analyzed in this study. **Cohort 3 comprised of 19 AYA patients from The Cancer Genome Atlas-Skin Cutaneous Melanoma (TCGA-SKCM) project, these patients do not have accessible clinical treatment record, their transcriptome data were used for the baseline comparison of immune profiles (CIBERSORT deconvolution) between AYA (total n = 47) and adult patients.**

8. On line 144, it says “Of the X AYA patients...” – what is X?

Response: We have corrected the typo.

- **RESULTS, page 5:** Of the **8** AYA patients...

9. The authors make the statement that ICI response rates were lower in AYA patients than adults. This is true when aggregating combination and monotherapy treated patients (responses are 38% in AYA and 63% in adults), but it seems like the data is actually much more nuanced. From my understanding of the breakdown of responses later in the paragraph discussing these results, it looks like 78% of AYA patients respond to combination treatment while 68% of adults do. I don't know if this ends up being a statistically significant difference that AYA patients respond better to combo ICI, but it is not at all consistent with the idea that ICI generally works worse for AYA patients. It seems like the conclusion that ICI responses are worse in AYA patients is driven only by responses monotherapy (not clear if it is true for both anti-CTLA-4 and anti-PD-1, or just one of those). This should be clarified in the text—the

monotherapy versus combo therapy dichotomy is a very interesting finding itself, but it does not seem appropriate to generalize to say all ICI is less effective in AYA patients, unless there is additional data to support his claim.

Response: We agree with the reviewer, and have made changes to the writing in the revised manuscript:

- **INTRODUCTION, page 3:** More recently, a retrospective study compared young adult melanoma patients ≤ 40 years to older age groups, and found no overall difference in immunotherapy response based on age, but a significantly higher response rate (53% vs 38%) and improved PFS (median 13.7 vs 4.0 months) with combination ICI compared to monotherapy in younger patients¹². The same study also analyzed patients ≤ 30 years, which also showed better or trends towards better outcomes with combination ICI vs anti-PD-1 monotherapy (objective response rates of 57% vs 44%, PFS 13.8 vs 4.0 months, OS 82.1 vs 24.1 months)¹².
- **RESULTS, page 5:** Overall, objective response to ICIs was lower among 21 AYA patients compared to 71 adult patients (38% vs 63%; Fisher-exact P = 0.047; Table 1). However, for AYA patients, combination ICI had a significantly higher response rate of 78% (7/9) compared to 5% (1/12) with monotherapy treatment (Fisher-exact P = 0.022).
- **DISCUSSION, page 18:** Similar to a previous study on young adult melanoma¹², we observed that AYA patients treated with combination ICI therapy had a higher response rate and longer progression-free survival than monotherapy...

Reference:

12. Wong SK, et al. Efficacy and safety of immune checkpoint inhibitors in young adults with metastatic melanoma. *European Journal of Cancer* (2023). <https://doi.org/10.1016/j.ejca.2022.12.013>

10. The images in Fig. 2 are very poor quality, and make it difficult to evaluate the staining. In addition to the panels shown, it would also be useful to see representative images for the Foxp3 channel for AYA and adult patients, illustrating the point that AYA melanomas typically had significantly more Tregs and supporting the quantifications shown in Fig. 2f. The Foxp3 insets shown make it look like the adult patient actually has more Tregs, although this is quite difficult to see with the current quality of the images.

Response: The representative staining images in Fig. 2d-e have been updated, as shown below.

Due to the limited file size that can be uploaded, the resolution was reduced. We have uploaded the high-resolution image to the Nature Manuscript Tracking System.

11. In Fig. 3c, the figure says “peritumor Tregs” while the text just says “Tregs” – this should be clarified which is being shown in Fig. 3c.

Response: The result in Fig. 3c is peritumour only, we have clarified this in the text:

- **RESULTS, page 9:** There was a non-significant, but numerically higher **peritumoural** T_{reg} density in the AYA ICI-resistant patients compared to the AYA non-resistant patients (Fig. 3c; median = 68 vs 102 cells/mm², P = 0.37).

If Fig. 3c is showing just peritumor Tregs, did the authors look at intratumoral Tregs as well?

We have added the intratumoral Treg results to Extended Data Fig. 3a, and described in the revised manuscript:

- Extended Data Fig. 3a

Extended Data Fig. 3 Spatial enrichment of Tregs is correlated with T cell dysfunction. (a) Comparison of intratumoural Treg density between ICI non-resistant and resistant AYA patients (cohort 1); P value of Mann-Whitney test is shown.

- **RESULTS, page 9:** T_{reg} density was not different between response groups in the intratumor region (Extended Data Fig. 3a).

12. In Extended Fig. 4c-d, there are p-values listed to the right of Fig. 4d. Are these p-values for the peritumoral cell densities (Fig. 4d) or also for the intratumoral cell densities (Fig. 4c)? A p-value for intratumoral densities should be included separately for Fig. 4c.

Response: We have updated the previous Extended Data Fig. 4c-d, now Extended Data Fig. 5a-b, to include non-resistant AYA patients (all patients in cohort 1) and show all P values, as shown below.

Extended Data Fig. 5 Distinct cellular profiles and survival outcomes in ICI non-resistant, SIL^{high} and SIL^{low} subtypes of AYA melanoma. (a) Intratumoural T cell density comparisons between ICI non-resistant and resistant Group 1 (SIL^{high}) and Group 2 (SIL^{low}) AYA patients; bar and lines represent mean and standard deviation; P values of Mann-Whitney tests are shown; NS, not significant ($P > 0.05$). **(b)** Peritumoural T cell density comparisons between ICI non-resistant and resistant Group 1 (SIL^{high}) and Group 2 (SIL^{low}) AYA patients; bar and lines represent mean and standard deviation; P values of Mann-Whitney tests are shown; NS, not significant ($P > 0.05$).

13. It would be nice to cell quantifications for non-resistant tumors included in the bar graphs in Extended Fig. 4c-d. Since this data is included in Fig. 3h for Gzmb+ CD8s, it seems the authors have collected this data, and it would be useful to see it included here and to be able to compare T cell infiltrates in non-resistant tumors to each of the two resistant subtypes. It would be really interesting to see whether “SIL-high” tumors (which have high Gzmb+ CD8s) generally have a T cell infiltrate similar to non-resistant tumors—this would be impactful, particularly if it can be coupled with data showing what fraction of non-resistant patients have tumors that look like SIL-high versus SIL-low resistant patients.

Response: We have added the AYA non-resistant patients into immunophenogram (Extended Data Fig. 4) and cell profile comparisons (Extended Data Fig. 5), updated figures as shown below, and added descriptions in text as detailed below.

- Extended Data Fig. 4

a Non-resistant

b SIL^{high}

c SIL^{low}

Extended Data Fig. 4 Immunophenograms of ICI non-resistant and resistant subtypes of AYA melanoma. (a) Immunophenograms of ICI non-resistant patients. (b) Immunophenograms of Group 1

(SIL^{high}) ICI-resistant patients. **(c)** Immunophenograms of Group 2 (SIL^{low}) ICI-resistant patients. Genes, category acronyms, and colour legends are shown in the example Immunophenogram at the bottom right. The +/- sign after the gene name represents the weighting when calculating the immunophenoscore. Abbreviations: MHC – Major histocompatibility complex (antigen processing); CP – Checkpoints (immunomodulators); EC – Effector cells; SC: Suppressor cells.

• Extended Data Fig. 5

Extended Data Fig. 5 Distinct cellular profiles and survival outcomes in ICI non-resistant, SIL^{high} and SIL^{low} subtypes of AYA melanoma. (a) Intratumoral T cell density comparisons between ICI non-resistant and resistant Group 1 (SIL^{high}) and Group 2 (SIL^{low}) AYA patients; bar and lines represent mean and standard deviation; P values of Mann-Whitney tests are shown; NS, not significant (P > 0.05). **(b)** Peritumoral T cell density comparisons between ICI non-resistant and resistant Group 1 (SIL^{high}) and Group 2 (SIL^{low}) AYA patients; bar and lines represent mean and standard deviation; P values of Mann-Whitney tests are shown; NS, not significant (P > 0.05). **(c)** Comparisons of CIBERSORT immune cell proportions between ICI non-resistant and resistant Group 1 and Group 2 AYA patients; bar and lines represent mean and standard deviation; P values of Mann-Whitney tests are shown; NS, not significant (P > 0.05). **(d-e)** Kaplan-Meier curves comparing the overall and progression-free survival between SIL^{high} and SIL^{low} subgroups of ICI resistant and non-resistant AYA patients. **(f-g)** Kaplan-Meier curves comparing the overall and progression-free survival between SIL^{high} and SIL^{low} groups of all AYA patients with recorded outcome.

- **RESULTS, page 11:** Group 1 melanomas were characterised by higher expression of effector cell, suppressor cell and checkpoint genes, **which reflected an overall similar profile compared to the non-resistant patients** (Fig. 3e, **Extended Data Fig. 4a-b**).
- **RESULTS, pages 11-12:** Multiplex IF was used to compare the spatial (peritumoural and intratumoural) densities of T cells in ICI non-resistant and resistant subgroups of AYA melanomas (**Extended Data Fig. 5a-b**, Supplementary Table 5). Intratumourally, the **non-resistant and SIL^{high} group (Group 1, high stroma infiltrating lymphocytes) had significantly higher densities of T_{reg} populations compared to the SIL^{low} group**...Peritumourally, the **non-resistant group and SIL^{high} group had significantly higher densities of multiple T_{reg} and tumour-specific CD39⁺ T cell populations compared to SIL^{low} group** (**Extended Data Fig. 5b**)...The **SIL^{high} ICI-resistant melanomas also had similar densities of intratumoural CD3⁺CD8⁺ cytotoxic T cells and higher densities of peritumoural CD3⁺CD8⁺ T cells (median = 1250 vs 623 cells/mm², P = 0.20) and CD3⁺PD1⁺ T cells (median = 508 vs 88.4 cells/mm², P = 0.036) compared to the ICI non-resistant group** (**Fig. 3h, Extended Data Fig. 5a-b**)... **non-resistant patients had higher M2 macrophage proportions compared to the SIL^{high} group** (median relative percentage = 12.3% vs 3.17%, P = 0.0021), and **higher plasma cell proportions compared to both SIL^{high} and SIL^{low} groups** (median relative percentage = 2.42% vs 0% and 0%, P = 0.013 and 0.012).

14. Text discussion of figures is not sequential, which is confusing. The text description of Extended Fig. 2e-f comes before Fig. 2c-d;

Response: We have revised the text to describe figures in sequence:

- **RESULTS, pages 6-7:** Comparison between the response groups showed that both ICI-resistant and non-resistant AYA melanomas harbored higher densities of T_{regs} compared to resistant (median = 72.8 and 122 vs 9.38 cells/mm², P = 0.0006 and 0.001) and non-resistant (median = 15.1 cells/mm², P = 0.0048 and 0.0069) adult melanomas (**Extended Data Fig. 2c**). The CD8⁺:FOXP3⁺ cell ratio showed a trend of difference between ICI-resistant and non-resistant adult patients (median = 2.18 vs 5.96, P = 0.058), but was not different between AYA resistant vs non-resistant groups (median = 2.74 vs 5.10, P = 0.77). In ICI-non-resistant AYA melanomas, the CD8⁺:FOXP3⁺ cell ratio was similar to the adult groups, while the resistant AYA group showed a trend of having lower CD8⁺:FOXP3⁺ cell ratio compared to adult non-resistant melanomas (P = 0.085) (**Extended Data Fig. 2d**). To determine whether T_{reg} cell density was different across age groups, we stratified the ICI-treated baseline AYA and adult melanoma patients into four age groups (15-30, 31-45, 46-60, 61-84) and assessed FOXP3⁺ T_{reg} cell densities. We found that the T_{reg} cell enrichment was only evident in the 15-30 (AYA) age group (Fig. 2h), with no difference in TME T_{reg} cell densities across older adult age groups (P = 0.33).

The transcriptomic T_{reg} proportion estimates were correlated with mIF T_{reg} quantification to determine the concordance of the approaches. In AYA melanomas, the expression of key gene markers for T_{reg} cells (*CD3D*, *FOXP3*) strongly correlated with CD3⁺FOXP3⁺ cell density (**Extended Data Fig. 2e**; Spearman's $\rho = 0.74$, $P < 0.0001$). Furthermore, the expression of functional T_{reg} genes (*CD3D*, *FOXP3*, *ICOS*) also correlated with its mIF (CD3⁺FOXP3⁺ICOS⁺ cell) density (**Extended Data Fig. 2f**; Spearman $\rho = 0.68$, $P = 0.00011$). The high proportion of T_{reg} cells confirmed the distinct TME profile of AYA melanomas.

and Fig. 3g comes before Fig. 3e-f

- **RESULTS, page 11:** The transcriptomic-based immunophenogram score²⁹ was used to evaluate the overall immunogenicity of ICI AYA melanomas (Extended Data Fig. 4), which qualitatively stratified the innately resistant tumors into high and low immunogenicity groups (**Fig. 3e-f**). Principal component analysis of transcriptomic profiling identified two subgroups of AYA ICI-resistant tumors (**Fig. 3g**).

REVIEWER 2 (expert in melanoma and adolescent melanoma)

This is an important study which demonstrated a peculiar immunogenomic profile in adolescent and young adult melanoma patients. Moreover, the study individualized specific tumor microenvironment features in immune checkpoint inhibitors-resistant adolescent and young adult melanoma.

The work is original, and the conclusions are consistent with the experimental data. The methodology is appropriate, and the study is very well structured.

Response: We thank the reviewer for supportive comments.

1. I would suggest including a treatment flow chart based on the transcriptomic, genomic, and spatial cellular profiles of the tumor microenvironment in adolescent and young adult melanoma.

We thank the review for the great suggestion, and have now included a treatment switch plan flow chart in Extended Data Fig. 6g based on the multi-omics analysis of our cohort 1 AYA patients.

g

Extended Data Fig. 6 Differentially expressed genes in AYA patients can help to predict response and inform treatment strategies in ICI-resistant patients. (g) Model treatment switch plan for ICI-resistant patients; bi-specific T cell engager (BiTE); adoptive cell transfer (ACT).

REVIEWER 3 (skin cancer and physiology, single-cell transcriptomics):

In this manuscript, Bai et al. compared the transcriptomic, genomic, and intratumoral immune landscapes of the tumor microenvironment (TME) of melanomas from adolescent and young adult (AYA) and adult patients to assess the pathways associated with immune checkpoint inhibitor (ICI) resistance in AYA patients, and to define biomarkers that allow treatment response prediction. Since the incidence of melanoma in young patients is significantly higher compared to older patients, and contradictory data about frequency of resistance to ICI treatment with age as a determining factor were reported, this manuscript addresses a timely question. The authors revealed that in general, response to ICI treatment and survival were poor among AYA patients treated with ICI. Their results suggest that AYA melanoma patients benefit from combination ICI treatment as far as response and survival are concerned. Furthermore, they identified 2 ICI-resistant subgroups of AYA melanoma which display differences in stroma infiltrating lymphocytes (SILs) compared to adult TME; SIL^{high} are enriched for Tregs (peritumorally) and present with alternative checkpoint molecule expression, while the SIL^{low} tumors showed a lack of immune induction. They also established an immunosuppressive score for AYA melanoma patients (YIM score) which predicted ICI response for the analyzed AYA cohort. They also proposed potential immune and non-immune based therapeutic targets for personalized therapies of ICI-resistant AYA patients. This is a descriptive study with most of the data generated through computational analysis of sequencing data. In general, the analysis pipeline of the datasets follows previously published computational approaches and seems to be well done. The authors performed CIBERSORT immune cell deconvolution on pre-treatment melanoma biopsy bulk-seq data to assess the tumor immune microenvironment, and confirmed some of their findings (increased Treg numbers in AYA patients, especially in peritumoral regions) by immunostainings. They also established a gene-expression-based immunosuppressive score for AYA melanoma patients (YIM score) which predicted ICI response for the analyzed AYA cohort. Furthermore they performed a drug gene interaction prediction analysis and suggested potential immune and non-immune based therapeutic targets for personalized therapies of ICI-resistant AYA patients.

Response: We thank the reviewer for positive comments and constructive feedback.

Major comments:

1. Fig. 2d-e: “we performed immunofluorescence (mIF) staining (Fig. 2d-e; Extended Data Fig. 2b) for Treg populations in the matched tumor”. What exactly is the matched tumor? I assume it is the pre-treatment biopsy but it is not clearly stated if it is before or after treatment?

Response: We have rephrased the writing in the revised manuscript to clarify on the samples used in the study:

- **RESULTS, page 6:** To further evaluate the T cell composition and spatial location in the TME, we performed multiplex immunofluorescence (mIF) staining (Fig. 2d-e; Extended Data Fig. 2b) on the **baseline tumor resections used for sequencing (cohort 1 and 2)**

Furthermore, cellular composition varies in different regions of a tumor. Was the whole tumor tissue used for quantification? If not, what were the selection criteria for the analyzed regions of interest?

Yes, the whole tumour tissue (tumour core and surrounding stromal region) was used for quantification and downstream analysis. We have clarified in the methods:

- **METHODS, page 24:** Image analysis section: **Whole tissue sections** were included for quantification and downstream analysis, **which included the tumor core and the immediate surrounding stromal (peritumor) region.**

2. Fig. 2/ Extended Figure 2: Are all AYA patients with high Foxp3 cell numbers resistant to ICI therapy?

Response: No, not all AYA patients with high FOXP3⁺ T_{regs} (cell density in mIF analysis) were resistant to ICI. Our initial observation was that a majority of ICI resistant (88%; 14 out of 16) and non-resistant (91%; 10 out of 11) AYA patients have higher T_{reg} densities compared to the median T_{reg} density in adult melanomas (12.99 cells/mm²). To provide clarity, we have rephrased the sentence in the updated manuscript to clarify that high T_{reg} is not a feature exclusive to resistant AYA:

- **RESULTS, page 6:** Comparison between the response groups showed that both ICI-resistant and non-resistant AYA melanomas harboured higher densities of T_{regs} compared to resistant (median = 72.8 and 122 vs 9.38 cells/mm², P = 0.0006 and 0.001) and non-resistant (median = 15.1 cells/mm², P = 0.0048 and 0.0069) adult melanomas (Extended Data Fig. 2c).

3. Is there a correlation between BRAF mutations and ICI resistance in AYA patients, or more specifically within SIL_{high} and SIL_{low} groups?

Response: BRAF mutation status was not associated with intra- or peritumoral T_{reg} densities in AYA patients, as shown in Extended Data Fig. 6a:

a CD3+CD8-FOXP3+ cell density

Extended Data Fig. 6 Differentially expressed genes in AYA patients can help to predict response and inform treatment strategies in ICI-resistant patients. (a) Comparison of peritumor Treg cell densities in AYA melanomas with and without the respective somatic mutations; bar plot shows mean and standard deviation.

According to Extended Fig. 7 most SIL_{low} patients have BRAF mutations while SIL_{high} have not. Do BRAF mutations correlate with Treg increases?

Response: We performed cell density analysis and Fisher's exact test (shown below, not added into the manuscript) to assess the association between *BRAF* mutation and T_{reg} infiltration or SIL spatiotype in ICI resistant AYA patients. Although 5 out of 6 SIL_{low} ICI resistant AYA patients were *BRAF* mutant, the mutation was not associated with changes in T_{reg} density, and was not associated with SIL spatiotype, in line with Extended Data Fig. 6a. This is possibly attributed to the small cohort size. The possible genotype association is out of the scope of the current study and we agree with the reviewer that it should be followed up in future studies. In addition, our GSEA analysis showed more tumour intrinsic pathways activated in SIL_{low} resistant patients compared to more extrinsic modulations in SIL_{high}, the association between genomic drivers and extrinsic features should be studied in a bigger cohort as we have already explained in the Discussion section (page 18) of the manuscript.

ICI resistant AYA patients (n = 16)

Table Analyzed	FisherExact_BRAFvsSILhi_SILlo_Resistant (n = 16)		
P value and statistical significance			
Test	Fisher's exact test		
P value	0.1451		
P value summary	ns		
One- or two-sided	Two-sided		
Statistically significant (P < 0.05)?	No		
Data analyzed	SIL^{high}	SIL^{low}	Total
BRAF mut	4	5	9
BRAF wt	6	1	7
Total	10	6	16
Percentage of row total	SIL^{high}	SIL^{low}	
BRAF mut	44.44%	55.56%	
BRAF wt	85.71%	14.29%	
Percentage of column total	SIL^{high}	SIL^{low}	
BRAF mut	40.00%	83.33%	
BRAF wt	60.00%	16.67%	
Percentage of grand total	SIL^{high}	SIL^{low}	
BRAF mut	25.00%	31.25%	
BRAF wt	37.50%	6.25%	

Table Analyzed	FisherExact_BRAFvsSILhi_SILlo_allAYA (n = 27)		
P value and statistical significance			
Test	Fisher's exact test		
P value	0.0912		
P value summary	ns		
One- or two-sided	Two-sided		
Statistically significant (P < 0.05)?	No		
Data analyzed	SIL^{high}	SIL^{low}	Total
BRAF mut	9	9	18
BRAF wt	8	1	9
Total	17	10	27
Percentage of row total	SIL^{high}	SIL^{low}	
BRAF mut	50.00%	50.00%	
BRAF wt	88.89%	11.11%	
Percentage of column total	SIL^{high}	SIL^{low}	
BRAF mut	52.94%	90.00%	
BRAF wt	47.06%	10.00%	
Percentage of grand total	SIL^{high}	SIL^{low}	
BRAF mut	33.33%	33.33%	
BRAF wt	29.63%	3.70%	

4. Fig. 3c/ line 240-242: “The density of Treg cells were higher in the AYA ICI-resistant patients compared to the AYA non-resistant patients”. There is a trend but no significant difference. Please rephrase!

Response: We have rephrased the sentence in the revised manuscript (as shown below).

- **RESULTS, page 9:** There was a non-significant, but numerically higher **peritumoural** T_{reg} density in the AYA ICI-resistant patients compared to the AYA non-resistant patients (Fig. 3c; median = 68 vs 102 cells/mm², P = 0.37).

5. Fig. 3: Is there a SIL^{low} group with a general lack of immune induction in the TME of adult resistant melanomas?

Response: We thank the reviewer for the interest in our data. In the current study, we could not apply the same SIL characterisation in the adult resistant melanomas due to differences in sample processing (the tumour region was prioritised for imaging, different algorithms used to classify tumour vs peritumour regions) which introduced confounding factors that would impact the results. We therefore did not want to include this in the manuscript as further standardised studies are warranted before any meaningful conclusions could be reached for the adult melanoma.

TME of adult melanomas has been widely characterised where the immune inflamed, excluded, and desert phenotypes have been well discussed (Hedge et al., 2020), with several studies from our group characterising multi-omics phenotypic features associated with ICI response and resistance in adult melanoma (Gide et al, 2019; Adegoke et al.,2023; Newell et al., 2022).

Potentially, there is a SIL^{low} group with a general lack of immune induction in adults. In our previous study of the adult melanoma cohort (Gide et al, 2019), we found significantly higher peritumoral CD8 and PD-1 cell densities at baseline in responders compared with non-responders to ICIs. The peritumoral FOXP3 cell densities were also higher in responders compared with non-responders early during treatment in combined immunotherapy. These findings showed increased lymphocyte trafficking to the tumour in responders as the intratumoural T cell densities generally correlated with the peritumoral densities. In another study, the co-authors have found that the presence of tertiary lymphoid structures improve immunotherapy outcome and survival in advanced melanoma (Cabrita et al., 2020).

References:

- Hegde, P. S., & Chen, D. S. (2020). Top 10 Challenges in Cancer Immunotherapy. *Immunity*, 52(1), 17–35. <https://doi.org/10.1016/j.immuni.2019.12.011>
- Gide, T. N., Quek, C., Menzies, A. M., Tasker, A. T., Shang, P., Holst, J., Madore, J., Lim, S. Y., Velickovic, R., Wongchenko, M., Yan, Y., Lo, S., Carlino, M. S., Guminski, A., Saw, R. P. M., Pang, A., McGuire, H. M., Palendira, U., Thompson, J. F., Rizos, H., ... Wilmott, J. S. (2019). Distinct Immune Cell Populations Define Response to Anti-PD-1 Monotherapy and Anti-PD-

1/Anti-CTLA-4 Combined Therapy. *Cancer cell*, 35(2), 238–255.e6. <https://doi.org/10.1016/j.ccell.2019.01.003>

- Adegoke, N. A., Gide, T. N., Mao, Y., Quek, C., Patrick, E., Carlino, M. S., Lo, S. N., Menzies, A. M., Pires da Silva, I., Vergara, I. A., Long, G., Scolyer, R. A., & Wilmott, J. S. (2023). Classification of the tumor immune microenvironment and associations with outcomes in patients with metastatic melanoma treated with immunotherapies. *Journal for immunotherapy of cancer*, 11(10), e007144. <https://doi.org/10.1136/jitc-2023-007144>
- Newell, F., Pires da Silva, I., Johansson, P. A., Menzies, A. M., Wilmott, J. S., Addala, V., Carlino, M. S., Rizos, H., Nones, K., Edwards, J. J., Lakis, V., Kazakoff, S. H., Mukhopadhyay, P., Ferguson, P. M., Leonard, C., Koufariotis, L. T., Wood, S., Blank, C. U., Thompson, J. F., Spillane, A. J., ... Long, G. V. (2022). Multiomic profiling of checkpoint inhibitor-treated melanoma: Identifying predictors of response and resistance, and markers of biological discordance. *Cancer cell*, 40(1), 88–102.e7. <https://doi.org/10.1016/j.ccell.2021.11.012>
- Cabrita, R., Lauss, M., Sanna, A., Donia, M., Skaarup Larsen, M., Mitra, S., Johansson, I., Phung, B., Harbst, K., Vallon-Christersson, J., van Schoiack, A., Lövgren, K., Warren, S., Jirström, K., Olsson, H., Pietras, K., Ingvar, C., Isaksson, K., Schadendorf, D., Schmidt, H., ... Jönsson, G. (2020). Tertiary lymphoid structures improve immunotherapy and survival in melanoma. *Nature*, 577(7791), 561–565. <https://doi.org/10.1038/s41586-019-1914-8>

6. Line 406/407: “Our study showed that combination ICI therapy had better progression-free survival than monotherapy in AYA melanoma patients, particularly those with Treg enrichment”. Where is the last part shown?

Response: We have re-worded the writing in the revised manuscript (as shown below).

- **DISCUSSION, page 18:** Similar to a previous study on young adult melanoma¹², we observed that AYA patients treated with combination ICI therapy had a higher response rate and longer progression-free survival than monotherapy, which may be associated with T_{reg} enrichment in the TME. These findings support the use of combination therapy as a potential strategy to overcome T_{reg}-mediated immunosuppression and improve treatment outcomes in AYA melanoma patients.

The response rates to immunotherapy were described in the results, and survival analysis comparing different treatments in AYA cohort 1 is shown in Extended Data Fig.1f:

- **RESULTS, page 5:** However, for AYA patients, combination ICI therapy had a significantly higher response rate of 78% (7/9) compared to 5% (1/12) with monotherapy treatment (Fisher-exact P = 0.022).
- Extended Data Fig. 1f:

Extended Data Fig. 1 Comparisons of overall and progression free survival in patients treated with ICI. Kaplan-Meier curves comparing the overall and (f) progression-free survival between AYA patients treated with anti-PD-1 (n = 10), anti-CTLA-4 (n = 5) and combination (Combi, anti-PD-1+anti-CTLA-4; n = 12) ICI.

7. Apart from higher proportions of Tregs in AYA melanomas, and higher numbers of naïve B cells and lower numbers of plasma cells and M2 macrophages were reported in AYA melanomas. Is there a correlation between these cell types with ICI resistance or the SILhigh and SILlow group?

Response: We have now added the comparison in Extended Data Fig. 5c and description of results in the main text:

- Extended Data Fig. 5c:

Extended Data Fig. 5 Distinct cellular profiles and survival outcomes in ICI non-resistant, SIL^{high} and SIL^{low} subtypes of AYA melanoma. (c) Comparisons of CIBERSORT immune cell proportions between ICI non-resistant and resistant Group 1 and Group 2 AYA patients; bar and lines represent mean and standard deviation; P values of Mann-Whitney tests are shown; NS, not significant ($P > 0.05$).

- RESULTS, page 12:** Since M2 macrophages and plasma cells were identified by the CIBERSORT analysis as significantly decreased in AYA melanoma compared to adults, and naïve B cells significantly increased, we also compared the estimated cell proportions between ICI non-resistant and resistant groups of AYA patients (Extended Data Fig. 5c). Of note, non-resistant patients had higher M2 macrophage proportions compared to SIL^{high} group (median relative percentage = 12.3% vs 3.17%, $P = 0.0021$), and higher plasma cell proportions compared to both SIL^{high} and SIL^{low} groups (median relative percentage = 2.42% vs 0% and 0%, $P = 0.013$ and 0.012). The distinct cell proportions could be indicative of immune regulation in the TME, but requires further imaging experiments to confirm the phenotype and spatial location of these cells.

8. Fig. 4c: no one can read the genes listed in the heatmap. It would make sense to only show the top 20 up- or downregulated genes or to zoom into some regions with genes of interest

Response: Figure 4c (now 4b) has been revised to highlight only the top 50 differentially expressed genes as shown below. We have also included the full gene list as Supplementary Table 6, which is attached to the Nature Manuscript Tracking System.

Fig. 4: Predicting immunotherapy resistance with the young immunosuppressed melanoma (YIM) score and drug target identification to overcome therapy resistance. (b) Heatmap of the top 50 differentially expressed genes between Group 1 and Group 2 ICI-resistant AYA patients which are correlated with KEGG pathways; top genes with adjusted-p < 0.05 were ranked by absolute fold change.

9. Fig. 4: Was the YIM score calculated from AYA patients from cohort 1 and 3?

Response: No, YIM score was calculated for cohort 1 due to the availability of clinical treatment and patients' response data.

If the score was calculated only with cohort 1, would the prediction fit to cohort 3 as a validation?

The clinical treatment and response data was not recorded or publicly inaccessible for cohort 3. The YIM score prediction was established to separate AYA ICI responders and non-responders. We acknowledge that proper validation of the YIM score requires a larger and independent dataset. However, it is very difficult to source well-curated immunotherapy treated AYA melanoma samples/datasets, and the current cohort is too small to be divided into different cohorts (i.e. discovery, training and testing). Given the small sample size, we have approached the YIM score using a rank-

based scoring algorithm that is independent of the sample size, and this scoring algorithm has been validated by our group (Mao et al., 2023) and others (Bhuva et al., 2019). We have now revised the Introduction, Results and Discussion sections to provide clarity, and to highlight the novel and preliminary nature of the current YIM score (as shown below):

- **INTRODUCTION, page 4:** We toned down the predictive value of the YIM score – “We identified the genomic, transcriptomic, and spatial cellular features of two major subtypes of ICI-resistant AYA tumors (stroma infiltrating lymphocyte (SIL) high and low tumor subtypes) and established an **immunosuppressive score for AYA melanoma patients which differentiated the ICI response in this cohort.**”
- **RESULTS, page 16:** We have now added “Given the limited cohort, we used a rank-based gene scoring approach that is independent of sample size”.
- **DISCUSSION, page 19:** We have rephased sentences to reflect the premature nature of the YIM score and discuss its limitations “Given the limited number of AYA patients who achieved durable ICI response in our study cohort, the non-resistant comparison group data could be influenced by features of acquired resistance. This was indicated by the similar immunophenogram profiles and comparable levels of T cell infiltration in the TME and between the non-resistant and SIL^{high} patients. Further studies of well curated ICI responsive AYA patients are needed to tease out the nuance of resistant groups and mechanisms of ICI response...We recognize that the predictive value of the YIM score requires validation in independent cohorts.”

Although cohort 3 (TCGA AYA patients) has no accessible clinical data of treatment to validate the prediction of immunotherapy outcome, we calculated the YIM score for cohort 3 and stratified them into YIM high and low groups based on median cut-off (Supplementary Table 8). We found a trend of better disease specific survival for patients with lower YIM scores (less immunosuppression), the below figure is added into the revised manuscript as Extended Data Fig. 6f:

- **RESULTS, page 16:** Lower YIM score further showed a trend of longer disease specific survival in TCGA AYA (cohort 3) patients (Extended Data Fig. 6f).

Extended Data Fig. 6 Differentially expressed genes in AYA patients can help to predict response and inform treatment strategies in ICI-resistant patients. (f) Kaplan-Meier curves comparing the disease-free survival of YIM score high vs low (using group median score as cut-off) TCGA AYA patients (cohort 3).

References:

- Mao, Y., Gide, T. N., Adegoke, N. A., Quek, C., Maher, N., Potter, A., Patrick, E., Saw, R. P. M., Thompson, J. F., Spillane, A. J., Shannon, K. F., Carlino, M. S., Lo, S. N., Menzies, A. M., da Silva, I. P., Long, G. V., Scolyer, R. A., & Wilmott, J. S. (2023). Cross-platform comparison of immune signatures in immunotherapy-treated patients with advanced melanoma using a rank-based scoring approach. *Journal of translational medicine*, 21(1), 257. <https://doi.org/10.1186/s12967-023-04092-9>
- Bhuvana, D. D., Foroutan, M., Xie, Y., Lyu, R., Cursons, J., & Davis, M. J. (2019). Using singscore to predict mutation status in acute myeloid leukemia from transcriptomic signatures. *F1000Research*, 8, 776. <https://doi.org/10.12688/f1000research.19236.3>

10. Taking into account that you find SIL^{high} and SIL^{low} groups within the AYA patients, would it make sense to calculate 2 separate YIM scores for each group?

Response: We established YIM scoring approach for predicting ICI response for AYA patients. In this work, we established a proof-of-concept that YIM scores can be potentially used to stratify resistance SIL^{high} and SIL^{low} groups and non-resistance groups within the AYA patients. Therefore, we do not think that 2 separate YIM scores should be calculated for different SIL groups.

Furthermore, could the SIL-low score also predict ICI resistance in adult melanoma patients?

No. This is because YIM was proposed as an AYA-specific gene score. We highlighted in the manuscript (page 16) that it has less than 1% of genes which overlap with published predictive gene signatures for adult melanoma, and we cross validated its lack of predictive power in adult published cohorts in Extended Data Figure 7.

Minor comments:

11. The labels of some figures are very small and difficult to read, e.g. Figure 3e,f,

Response: We have increased the size of the labels to fit within the page limit in the revised manuscript. For example, Figures 3e and 3f (as shown below).

Fig. 3: Resistant hotspot of peritumoral Tregs and immunosuppressed spatiotypes of AYA melanoma at pre-treatment baseline. (e-f) Immunophenogram gene expression of AYA ICI-resistant patients; representative immune expression profiles of Group1 (e; n = 10) and Group 2 (f; n = 6) patients. Abbreviations: MHC – Major histocompatibility complex (antigen processing); CP – Checkpoints (immunomodulators); EC – Effector cells; SC: Suppressor cells.

Figure 4c and similar plots in the extended data.

For Figure 4c, which is now Figure 4b in the revised manuscript, has been revised to highlight only the top 50 differentially expressed genes. The full gene list is included as Supplementary Table 6. We have checked the extended data and have increased the size of the labels that fit within the page limit in the revised version of the manuscript.

12. Fig. 4: legends for b and c have been swapped;

Response: The labels have been revised in the updated manuscript (as shown below).

Fig. 4: Predicting immunotherapy resistance with the young immunosuppressed melanoma (YIM) score and drug target identification to overcome therapy resistance. (a) Heatmaps showing the gender, immunotherapy response for non-adjuvant (Adj.) treatment (complete response CR, partial response PR, progressive disease PD), immune phenotype (IP) group, gene signature scores, somatic variants, and mIF T cell densities (peritumoral, peri; intratumoral, tumo) of AYA melanomas (n = 28); not determined, ND. **(b)** Heatmap of the top 50 differentially expressed genes between Group 1 and Group 2 ICI-resistant AYA patients which are correlated with KEGG pathways; top genes with adjusted-p < 0.05 were ranked by absolute fold change. **(c)** Top pathways in KEGG gene set enrichment analysis (adjusted-P < 0.05). **(d)** Sankey diagram highlighting the alternative drug-gene interactions in AYA ICI

resistant patients (Supplementary Table 12); left represents the total number of druggable gene and mutational targets in SIL_{high} and SIL_{low} groups, middle shows targetable expressed or mutated markers, right shows the drug categories; line thickness represents the proportion within each group, for example, line will have a thickness of 1 if one patient expressed the drug target; bi-specific T cell engager, BiTE.

REVIEWERS' COMMENTS

Reviewer #1 (Remarks to the Author):

The revisions have substantially strengthened this manuscript. The authors' additional analysis of the non-resistant AYA patients in comparison to SIL-high and SIL-low resistant groups and the more nuanced language throughout regarding interpretation of results are in particular appreciated. My concerns have been addressed.

Reviewer #3 (Remarks to the Author):

In their revised manuscript, the authors have addressed all of my concerns, rephrased the text if their conclusions were not supported by data, and clarified critical open questions.